# Interindividual- and blood-correlated sweat phenylalanine multimodal analytical bio-chips for tracking exercise metabolism

Bowen Zhong[1,2], Xiaokun Qin[1,2], Hao Xu[1,2], Lingchen Liu[1,2], Linlin Li[1,2], Zhexin Li[1,2], Limin Cao[3], Zheng Lou [1,2], Joshua A. Jackman [4], Nam-Joon Cho[5] & Lili Wang [1,2] ✉

In situ monitoring of endogenous amino acid loss through sweat can provide physiological insights into health and metabolism. However, existing amino acid biosensors are unable to quantitatively assess metabolic status during exercise and are rarely used to establish blood-sweat correlations because they only detect a single concentration indicator and disregard sweat rate. Here, we present a wearable multimodal biochip integrated with advanced electro-chemical electrodes and multipurpose microfluidic channels that enables simultaneous quantification of multiple sweat indicators, including phenyla-lanine and chloride, as well as sweat rate. This combined measurement approach reveals a negative correlation between sweat phenylalanine levels and sweat rates among individuals, which further enables identification of individuals at high metabolic risk. By tracking phenylalanine fluctuations induced by protein intake during exercise and normalizing the concentration indicator by sweat rates to reduce interindividual variability, we demonstrate a reliable method to correlate and analyze sweat-blood phenylalanine levels for personal health monitoring.

Amino acids (AAs) are the building blocks of life and essential for the synthesis of proteins that play a critical role in human growth, maintenance, immunity, and reproduction[1–3]. In addition to protein-bound forms, free AAs are metabolic reactants and products that participate in a range of vital biological processes to maintain body function and homeostasis, including nutrition, metabolism, and physiological regulation[4–6]. Free AAs are present in all biofluids and tissues of the body, and their levels are associated with body status, such as exercise, diet, infection, disease, and psychology[7–12]. Thus, the analysis of free AAs in biofluids holds practical significance for assessing health conditions. Among various biofluids, the bio-marker utility of sweat AAs has received less attention except for

skin diseases because they arise from both the endogenous loss of plasma AAs[13] and the presence of AAs on the skin surface[14,15], which are referred to as natural moisturizing factors (NMF)[16] (Fig. 1a). The presence and contamination of NMFs in sweat has proven a critical challenge in the development of wearable sweat biosensors for AA monitoring[17–19]. Fortunately, recent studies have shown promising results indicating that the contribution of skin-leached AAs like NMFs to sweat is diminished and may eventually be exhausted with prolonged exercise duration, especially for AAs with low skin pre-valence (non-NMFs)[20–22]. Consequently, measured sweat AA con-centrations can be indicative of their blood levels and further serve as reliable biomarkers for assessing physiological status, which

[1]State Key Laboratory for Superlattices and Microstructures, Institute of Semiconductors, Chinese Academy of Sciences, Beijing 100083, China. [2]Center of Materials Science and Optoelectronic Engineering, University of Chinese Academy of Sciences, Beijing 100049, China. [3]Tianjin Key Laboratory of Lung Cancer Metastasis and Tumor Microenvironment, Tianjin Lung Cancer Institute, Tianjin Medical University General Hospital, Tianjin 300052, China. [4]School of Chemical Engineering and Translational Nanobioscience Research Center, Sungkyunkwan University, Suwon 16419, Republic of Korea. [5]School of Materials Science and Engineering, Nanyang Technological University, 637553 Singapore, Singapore. ✉e-mail: liliwang@semi.ac.cn

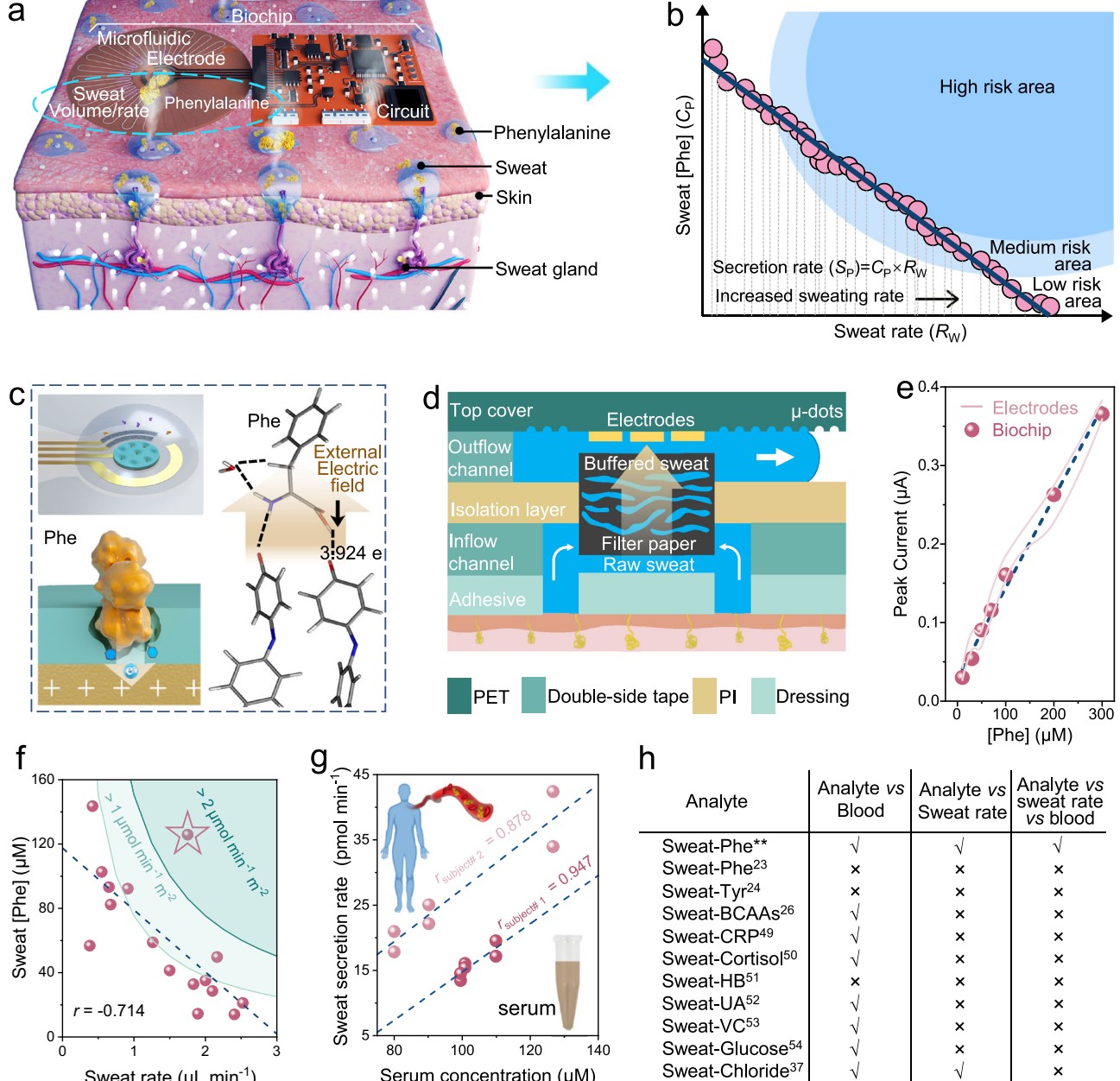

**Fig. 1 | Schematics of wearable multimodal biochip for assessing exercise metabolic risk and serum correlation through sweat analysis. a** Schematic of the biochip on skin for multimodal sweat sensing and two sources of sweat Phe, including skin surface and blood partitioning. **b** Indicative negative correlation between sweat Phe concentrations and sweat rates among individuals, along with the calculation method of Phe secretion rate (inlets area of 75.36 mm²) and the division of different metabolic risk areas. **c** Mechanism of the E-MIP electrode for detecting Phe by direct electrocatalytic oxidation and theoretical simulation of charge transfer between the E-MIP electrode and a Phe molecule under an external electric field. The red arrow represents the direction of the electric field. **d** Working principle and cross-section of the microfluidic module with vertically assembled structure. **e** Comparison of Phe DPV responses of the integrated wireless system and pristine electrodes. **f** Correlation of sweat rates and Phe levels among individuals at the same exercise time point, along with division of different sweat Phe secretion rates. The starred data indicates the subject with the highest secretion rate in this study. Data from 16 healthy subjects. **g** Correlations of sweat and serum Phe level in two subjects. The dashed lines represent linear-fitted trendlines. **h** Comparison of recent advances in sweat sensing. **: this work, Tyr tyrosine, BCAAs branched chain amino acids, CRP C-reactive protein, HB β-hydroxybutyrate, UA uric acid, VC Vitamin C.

provides attractive opportunities for non-invasive and convenient health monitoring.

Practically, wearable sweat AA biosensors are still at an early stage of progress, and existing biosensors can only detect a limited number of AAs in sweat[23–26]. There are still many AAs in sweat that remain to be explored and studied by wearable biosensors and hence applied to health practical scenarios according to their uncovered, potential blood correlations[27,28]. Phenylalanine (Phe), an essential AA, is present in sweat but has only a low presence on the skin surface (non-NMF) and

is hypothesized to partition into sweat via diffusion due to its small size and nonpolar side chain properties[14,29]. These physicochemical characteristics suggest that Phe would be useful as a sweat biomarker, especially since its sweat concentration could likely be correlated with blood Phe levels[30,31]. From a healthcare perspective, Phe concentration testing is utilized to assess various nutrition and disease statuses. For Phenylketonuria (PKU) patients, blood Phe testing is an important tool for early diagnosis in neonatal screening and ongoing dietary therapy throughout life[32]. Fluctuations in Phe concentrations are also linked to

muscle protein metabolism during exercise[33], liver dysfunction in obesity[34], and viral infection severity[11]. Importantly, there is a strong and direct correlation between Phe and overall AA levels in sweat[21], suggesting the potential to further incorporate sweat loss measurements to quantify sweat AA loss to assess metabolic status during exercise. In this regard, tracking exercise metabolism and assessing risk by monitoring sweat AA losses can avoid a net negative nitrogen balance due to depletion of free AAs resources, which leads to fatigue and pain during exercise[20,35]. Additionally, simultaneous measurement of Phe concentration and sweat rate can be used to determine the source of sweat Phe (i.e., secretion mechanism) at different stages of exercise sweating. By enabling an interindividual comparison with multiple human subjects, the obtained correlation between these two indicators can also be used to study the mechanism of sweat Phe partitioning from blood[36]. Understanding the mechanisms of sweat Phe partitioning and secretion paves the way for more sophisticated modeling to account for the sweat rate[37], which may produce a stronger relationship between sweat and blood Phe levels among individuals.

Recent efforts in the portable detection of blood or urine Phe concentrations mainly rely on molecular recognition elements (e.g., antibodies and aptamers), which are faced with challenges such as high cost, low stability, extensive washing requirements, and potential bioreactivity[38–41]. By contrast, electrochemical sensors that do not depend on bio-affinity interactions are widely considered more suitable to fulfill the design requirements of wearable devices, especially considering their low cost and ease of fabrication[42]. Since Phe is non-electroactive for common electrode materials such as gold, carbon, and graphene, there are two reported approaches for wearable electrochemical measurements to detect Phe, including electroactive derivatization of Phe and indirect detection using redox probes[23,43]. The former approach can only enable one-time Phe measurements[43], making it hardly adaptable for continuous and long-term wearable measurement. As for the latter, redox probes have been combined with molecularly imprinted polymers (MIP) to indirectly detect non-electroactive AAs[23,26], but the sensitivity of this method is relatively lower than the direct oxidation of analytes due to a nonlinear inverse relation between AA concentrations and peak current changes. Therefore, a wearable sensor for sweat Phe detection based on direct signal transduction has not yet been demonstrated (Supplementary Table 1). Moreover, simultaneous measurement of sweat AA levels and sweat rates as well as their combined analysis have also not been demonstrated and thus far disregarded in the field of wearable sweat sensing (Fig. 1h), due to lack of a convenient and reliable approach for quantifying sweat rate.

In this article, we present a wearable multimodal biochip for sensing multiple indicators, including sweat Phe and chloride concentrations, as well as sweat rate, which can together enable quantitative assessment of metabolic status during exercise. This biochip incorporates three functional modules for advanced in situ sweat detection (Fig. 1a): (i) an electrochemical electrode modified by an electrocatalytically active MIP for direct and selective determination of sweat Phe; (ii) a well-designed multipurpose microfluidic module that allows rapid sweat sampling, concentration refreshing, and pH buffering for a stable testing environment, as well as flow visualization for sweat loss measurement; and (iii) integration with a matching wireless flexible circuit and mobile software. Using this biochip approach, we investigated the variation in Phe levels between two human test subject groups with different body mass index (BMI) values, which can be attributed to the difference in sweat rates[20]. Leveraging this negative correlation, we analyzed the possible mechanism of Phe partitioning into sweat. Furthermore, we assessed exercise metabolic status and risk among volunteers with different physiological characteristics by using a composite indicator, the Phe secretion rate ($S_P$), derived from the sweat Phe concentration ($C_P$) and sweat rate ($R_W$) (Fig. 1b). Finally,

we demonstrate similar and strong correlations between sweat and serum Phe levels in different volunteers before and after protein intake via sweat rate normalization to reduce interindividual variability. All these demonstrations reveal the potential utility of our wearable multimodal system for sweat-based personalized exercise and diet management.

## Results

### Integrated system sensing strategy and applications

The biochip integrating multiple functional modules can be attached to the skin and samples the secreted sweat on the skin surface for multimodal sensing, data processing, and wireless transmission (Fig. 1a and Supplementary Fig. 1). All modules of the biochip can be prepared at a large scale through simple processing techniques, including thermal evaporation and laser engraving (Supplementary Figs. 2, 3).

To achieve high sensitivity and reliable detection of sweat Phe, we developed a Phe-imprinted MIP that mimics the functions of biological enzymes. Unlike regular MIPs that act as 'molecular filters' or 'artificial antibodies'[23], our MIP-based 'artificial enzyme' allows for direct and selective electrocatalytic oxidation of Phe on the electrode surface. This approach enables the direct electrochemical determination of sweat Phe concentration using differential pulse voltammetry (DPV). The enzyme-mimicking MIP can not only selectively bind with the target Phe in sweat through specific binding sites, but also directly electro-oxidize Phe via electrocatalytically active surface functional groups (Fig. 1c). The Phe-imprinted MIP electrode was synthesized by electropolymerizing polyaniline (PANI), which has been used for the chiral recognition of aromatic AAs (Tyr, Trp, and Phe) due to its own electrocatalytic activity[44–46].

Successful template imprinting and extraction from the electropolymerized PANI matrix were verified by various characterization methods (AFM, SEM, and FTIR-ATR) and molecular interaction simulations (Supplementary Note 1 and Supplementary Figs. 4–8). Furthermore, an improved theoretical intermolecular simulation based on density functional theory (DFT) calculations for studying the electrochemical behavior of Phe on the MIP electrode surface supports that the electrocatalytic phenomenon of the PANI-MIP electrode for detecting Phe in this system is related to the carbonyl (C = O) groups on the quinone rings of the polymer chains generated by the electro-degradation of PANI (Fig. 1c and Supplementary Note 2). Specifically, under the same external electric field, a greater charge transfer number ($\Delta q = 3.924e$) is calculated between the electro-degraded MIP electrode (E-MIP) and a Phe molecule compared to other electrode cases (Supplementary Fig. 9).

To obtain stable and reliable Phe electrochemical responses in complex sweat samples with variable pH and electrolyte concentration conditions, our biochip system integrates a vertically assembled microfluidic module with a neutral pH buffer filter paper embedded in the sensing chamber (Fig. 1d). As sweat flows through the filter paper loaded with dried phosphate buffer (PB, pH 7.5, 20×), this embedded design maintains the solution environment at a constant pH and high ionic strength level to achieve stable Phe sensing responses. Notably, as shown in Fig. 1e, the introduction of filter paper in front of the electrodes had negligible effects on the DPV responses of the integrated Phe sensing system within the normal physiological concentration range compared to that of pristine electrodes in PBS. In addition, by incorporating the design of a serpentine outflow channel with a rough upper surface, the microfluidic module also allows visualization of sweat flow for sweat loss quantification, including sweat volume and rate (Supplementary Note 3).

We explored the feasibility of combining the measured bimodal signals (sweat Phe concentration and sweat rate) to analyze the Phe partitioning mechanism into sweat, assess exercise metabolic status, and investigate the relationship between sweat and serum Phe levels (Supplementary Fig. 10). In detail, by using the wearable system to

collect data from multiple volunteers after 20 min of exercise, we observed a moderately negative correlation between sweat rate and Phe concentration (Fig. 1f), indicating that the mechanism of Phe partitioning into sweat relies on diffusion and its concentration may be affected by sweat dilution[47]. Remarkably, the correlation in the low sweat rate region was poorer than that in the high sweat rate region because the skin surface Phe content in sweat, which is affected by individual skin quality differences, was difficult to be exhausted by perspiration at low sweat rates[20]. Moreover, we also defined an ideal indicator for assessing exercise metabolic status with reduced inter-individual variability, i.e., sweat Phe secretion rate ($\mu mol\ min^{-1}\ m^{-2}$), and identified an individual with possible high metabolic risk during exercise whose secretion rate was greater than that of other volunteers (Fig. 1f). Here, we divided Fig. 1f into three metabolic risk areas, where lower than $1\ \mu mol\ min^{-1}\ m^{-2}$ or higher than $2\ \mu mol\ min^{-1}\ m^{-2}$ were defined as low risk or high risk, respectively, and between the two values was defined as medium risk. The division was based on a combination of actually measured values from subjects in this study and the similar sports science research[20]. Finally, through normalizing sweat indicator concentrations by sweat rates to reduce inter-individual variability, similar correlations between sweat and serum Phe levels were observed in two different subjects (Fig. 1g), indicating the potential of sweat Phe quantification for non-invasive personalized healthcare management. While there have been several recent pioneering and significant advances in sweat AA sensing[23,24,26], an in-depth investigation of AA partitioning mechanisms in conjunction with sweat rate detection and normalization has not yet been conducted due to the lack of multimodal analysis based on supplemental sweat rate measurements in parallel (Fig. 1h). Moreover, the use of other commonly measured sweat biomarkers and their analysis in combination with sweat rate measurements for health monitoring have also been lacking[24,36,48–54].

## Electrochemical characterization of E-MIP sensor

The E-MIP electrode showed superior electrocatalytic oxidation for Phe over the electro-degraded non-imprinted PANI (E-NIP) electrode due to fewer binding sites on the E-NIP electrode for Phe to access (Fig. 2a; see also Supplementary Note 1 and Supplementary Fig. 11). In addition, the gold electrode and polypyrrole (PPY) based MIP electrode had negligible responses to Phe due to the lack of effective functional groups. These experimental results indicate that the electro-oxidation of Phe occurs more readily on the E-MIP electrode than other tested electrodes. These results agree with theoretical intermolecular simulations (DFT) used to calculate the electron transfer number between electrodes and Phe molecules under an external electric field (Fig. 2b, c and Supplementary Fig. 9). The incorporation of specific binding sites and active functional groups within the E-MIP enabled the direct, selective, and sensitive electrocatalytic oxidation of Phe (Supplementary Table 2).

A well-defined increase in peak current readouts could be detected by DPV scans in the presence of increasing Phe concentrations from 10 to 1500 μM (Fig. 2d). Two linear relationships were determined: (i) from 10 to 300 μM, which had a sensitivity of $1.4\ nA\ \mu M^{-1}$ and a limit of detection (LOD) of 4.7 μM; and (ii) from 300 to 1000 μM, which had a sensitivity of $0.27\ nA\ \mu M^{-1}$ (Fig. 2e). As a control, a comparative experiment using the E-NIP electrode demonstrated appreciably smaller responses to similar Phe concentrations (Fig. 2f). This difference between MIP and NIP was also characterized in $[Fe(CN)_6]^{3-/4-}$ solution by cyclic voltammograms (CV) and electrochemical impedance spectroscopy (EIS) (Fig. 2g and Supplementary Fig. 12). Notably, EIS measured in 200 μM Phe further confirmed that the E-MIP electrode is a more suitable material for electro-oxidizing Phe because the fitting slope (−0.88) referring to Warburg impendence is close to −1 (Fig. 2h and Supplementary Note 1).

The E-MIP-based Phe sensor showed selective recognition of the Phe target, and effective discrimination against other AAs in sweat at physiologically relevant high concentrations (Fig. 2i), along with a variety of common sweat interferents (Fig. 2j and Supplementary Fig. 13). Meanwhile, it also exhibited the chiral recognition selectivity for the L enantiomer (L-Phe) compared to the D enantiomer (D-Phe) due to the use of L-Phe as the imprinted template (Supplementary Fig. 13). The all electro-processed MIP layer could be formed scalably on evaporated gold electrodes, which resulted in high reproducibility in terms of batch-to-batch consistency and continuous successive measurement stability (Fig. 2k and Supplementary Figs. 14, 15). The effect of pH changes on the responses of the Phe sensor was evaluated from pH 6 to pH 11, which indicated a relatively stable response between pH 7.0 and pH 9.5 (Fig. 2l, Supplementary Fig. 16, and Supplementary Note 1). However, human sweat is weakly acidic (pH 5 to pH 7), which motivated us to incorporate a microfluidic method for neutral pH buffering to accurately measure sweat Phe in our biochip system.

## Design and performance characterization of multipurpose microfluidics

The multipurpose microfluidic module for wearable sweat Phe sensors was fabricated by laser-engraving, which involved the patterning of channels and filter papers as well as surface roughening (see Methods for details). Each functional layer of microfluidics was integrated in a vertical direction, rendering a delicate 3D structure (Fig. 3a). The resulting microfluidics were capable of not only improving sweat sampling with higher temporal resolution for sweat sensing, but also serving the following purposes: 1) inlet and chamber optimization for fast sweat collection and refreshing; 2) visualized and serpentine outflow channels for in situ assessment of sweat loss status; and 3) maintenance of a pH-neutral and high ionic strength solution environment for accurate sweat Phe detection.

First, the combination of an increased number of inlets and an elliptic inlet design instead of a circle one increased the sampling area to collect sweat. Simultaneously, the presence of filter paper within the sensing chamber also decreased the chamber volume that needed to be filled with collected sweat (Supplementary Note 3 and Supplementary Fig. 17). With an experimentally measured sweat rate ($2\ \mu L\ min^{-1}$) as the inlet flow rate, the synergy of these two structural optimizations achieved rapid filling of the sensing chamber in the case of twelve inlets (around 8 min from starting exercise) (Fig. 3b). Finite element analysis (FEA) of the two-phase water/air filling process (Fig. 3c, Supplementary Fig. 18, and Supplementary Video 1) shows accordance with experimental time-lapse images of sweat filling in a volunteer during exercise. Furthermore, numerical simulations also showed that the embedding of filter paper can accelerate and homogenize the refreshing process, which is referred to as the refreshing time taken for the old solute concentration in the chamber to adjust to a new concentration during sweat inflow (Fig. 3d, e, Supplementary Fig. 19, and Supplementary Video 2).

Second, an upper three-layer was introduced into the vertical microfluidic structure to obtain adequate outflow channel volume for readable sweat loss while controlling the overall device footprint (Supplementary Note 3 and Supplementary Fig. 20). In brief, the differential reflectivity/transmittance to visible light at μ-dots leads to an apparent visual color change of the serpentine outflow channel (one meander corresponding to 1 μL, volume resolution of 0.5 μL) in empty and water-filled states[55,56] (Fig. 3f, g and Supplementary Figs. 21, 22), which enables visualized flow readouts for estimating sweat rate/volume on the skin. To validate this capacity, a physiological range of constant flow rates (from 0.5 to $2\ \mu L\ min^{-1}$) was injected into the microfluidics by syringe pump. The readable fluid filling positions over time were converted to flow rates measured by the naked eye, showing correspondence between injected and measured flow rates (Fig. 3h

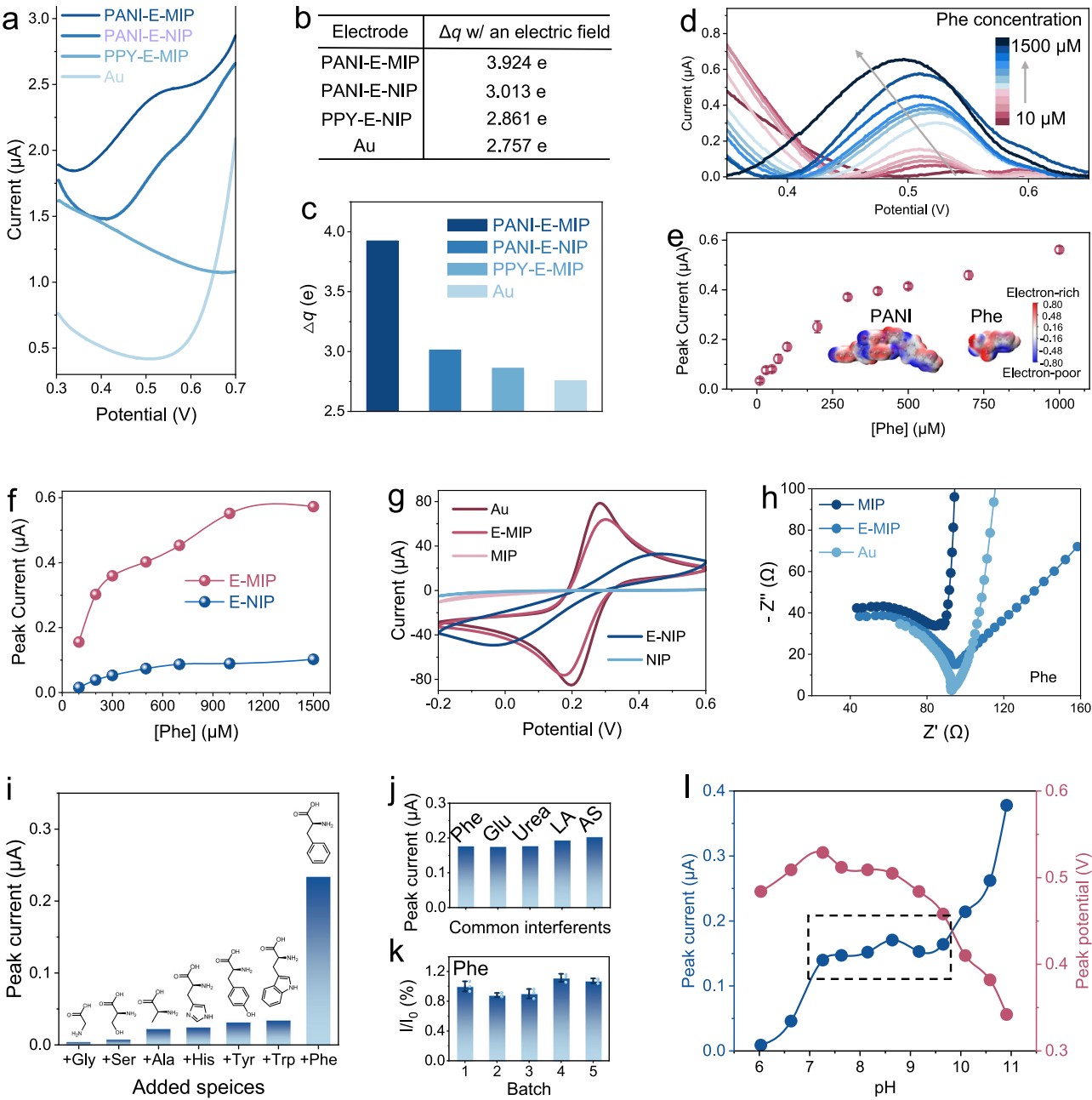

**Fig. 2 | Characterizations of E-MIP sensor for Phe detection. a** DPV scans of a PANI-based E-MIP electrode (PANI-E-MIP), a PANI-based E-NIP electrode (PANI-E-NIP), a PPY-based E-MIP electrode (PPY-E-MIP) electrode, and a gold electrode (Au) in a 10× PBS containing 200 μM Phe. **b, c** Comparison of the electron transfer number between different electrodes and Phe molecule within an external electric field. **d, e** DPV scans of the E-MIP-based Phe sensor for direct Phe detection after baseline correction (**d**) and corresponding peak current readouts (**e**). Inset, the molecular electrostatic potential surfaces of the E-MIP electrode and Phe. The error bars ($n = 3$ measurements) correspond to the standard deviation (SD). **f** Response comparison between E-MIP and E-NIP electrodes to equivalent Phe concentrations in PBS. **g** CV scans of different MIP electrodes based on PANI and an Au electrode in a solution containing 5 mM $[Fe(CN)_6]^{3-}$ and 0.2 M KCl. **h** EIS responses of an E-MIP electrode, a MIP electrode, and an Au electrode in a PBS containing 200 μM Phe. **i** Selectivity of the E-MIP sensor against other AAs. The following substances were added in succession: 1 mM Glycine (Gly), 1 mM Serine (Ser), 500 μM Alanine (Ala), 500 μM Histidine (His), 200 μM Tyr, 200 μM tryptophan (Trp), and 200 μM Phe. **j** Selectivity of the E-MIP sensor against common sweat interferents in presence of 200 μM Phe. The following interferents were added in succession: 100 μM glucose (Glu), 5 mM Urea, 5 mM lactate (LA), and 100 μM ascorbic acid (AS). **k** Batch-to-batch variation of the E-MIP sensor performance in the presence of 200 μM Phe. The error bars correspond to the standard deviation ($n = 3$ independent sensors). The center for the error bars represents the mean value. **l** Influence of pH changes in peak currents and respective peak potentials. The dashed box indicates the pH range with stable DPV responses.

and Supplementary Fig. 23). Additionally, a computer vision algorithm was used for automatic sweat loss reading (Supplementary Note 3). Based on this, the exercise sweat rates of two identical body parts (forehead and forearm) of eight volunteers were measured to demonstrate the practicality of this visualization design (Fig. 3i, Supplementary Fig. 24, and Supplementary Table 3), which can be used to

comprehensively investigate the relationship between AA loss and water loss during exercise.

Third, a disposable and replaceable neutral pH-buffering filter paper was laser-cut and embedded within the sensing chamber of the microfluidics. The injected 80 μL lactate acid (LA) solution (pH 5.0) was continuously buffered to around pH 7.0 (Fig. 3j), showing the

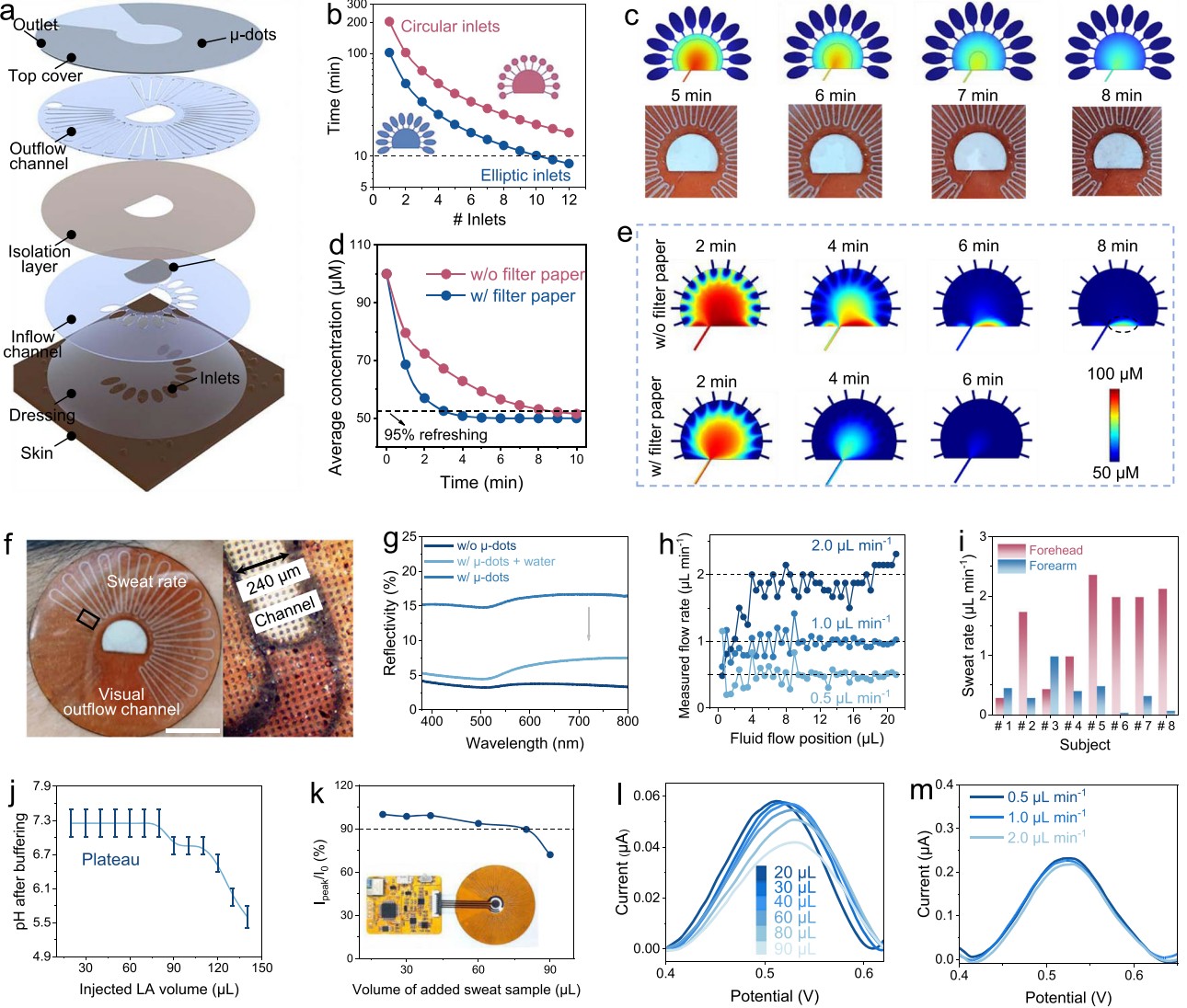

**Fig. 3 | Design and characterizations of multipurpose microfluidics for sweat sampling. a** Layer-by-layer view of microfluidic device design. **b** Numerical stimulation of time required to fill chamber for different numbers and shapes of inlets with embedded filter paper. **c** Comparison between simulated and experimental results of the sweat sampling/filling process. **d** Time evolution of the average Phe concentration for refreshing process in the chamber without or with embedded filter paper. **e** FEA of microfluidic refreshing process without or with embedded filter paper. The area marked by the dashed line is represented as an anomalous hard-to-refreshing area. **f** Photographs of the microfluidics during exercise on the skin (right) and optical micrograph of sweat flowing in the visualized microchannel (left, magnified view of the sweat front). Scale bar, 1 cm. **g** Optical reflectivity of empty channels with or without µ-dots and filled channels with µ-dots. **h** Measured flow rates by naked eye at different fluid filling positions under pump injection rates of 0.5, 1, or 2 µL min⁻¹. **i** Sweat rates measured by the algorithm form two body parts of eight healthy subjects during 10 to 20 min of exercise. **j** Neutral pH buffering capability of the embedded filter paper in the microfluidic chamber. Note: there were no error bars here, but the pH ranges measured by colorimetric pH test papers. **k, l** Sensor response changes (**k**) and corresponding DPV scans (**l**) caused by injection of different sweat sample volumes. Inset, photograph of the biochip. **m** DPV scans of the integrated wireless system at different flow rates (from 0.5 to 2 µL min⁻¹).

ability to buffer sweat to neutral pH conditions for more than 40 min during intense exercise with a persistently high sweat rate of 2 µL min⁻¹. After incorporating the multipurpose microfluidics for sweat sampling and buffering as well as a matching flexible circuit for wearable DPV measurement (Fig. 3k, inset), wireless and continuous Phe sensing of the multimodal sensors was validated via injection of successive target solutions (raw sweat samples) at physiological sweat rates (Fig. 3k–m and Supplementary Fig. 25). When continuously injecting sweat samples, the sensor performance maintained a stable response (less than 10% variation) with the first 80 µL of sweat injection (Fig. 3k, l), corresponding to the aforementioned neutral pH buffering capability of the microfluidics. The results shown that the embedded pH-buffering filter paper in microfluidics can effectively alleviate the influence of pH as well as ionic strength changes on the

sensor response within the design expectation. Furthermore, A stable response of the sensor was been obtained in the skin physiological temperature range (Supplementary Fig. 26). More importantly, the integrated sensor also exhibited stable DPV readouts for Phe sensing at different flow rates from 0.5 to 2 µL min⁻¹ (Fig. 3m), revealing that changes in sweat rate had a negligible effect on sensor responses. We also integrated an ion-selective Cl⁻ sensor into this multimodal system for more comprehensive exercise monitoring and management (Supplementary Fig. 27).

### Sensor evaluation for assessing exercise metabolic risk

The integrated multimodal sweat sensor could be suitably worn for in situ measurement and display of sweat Phe, chloride, and sweat loss levels during exercise (Fig. 4a–c, Supplementary Video 3, and Video 4).

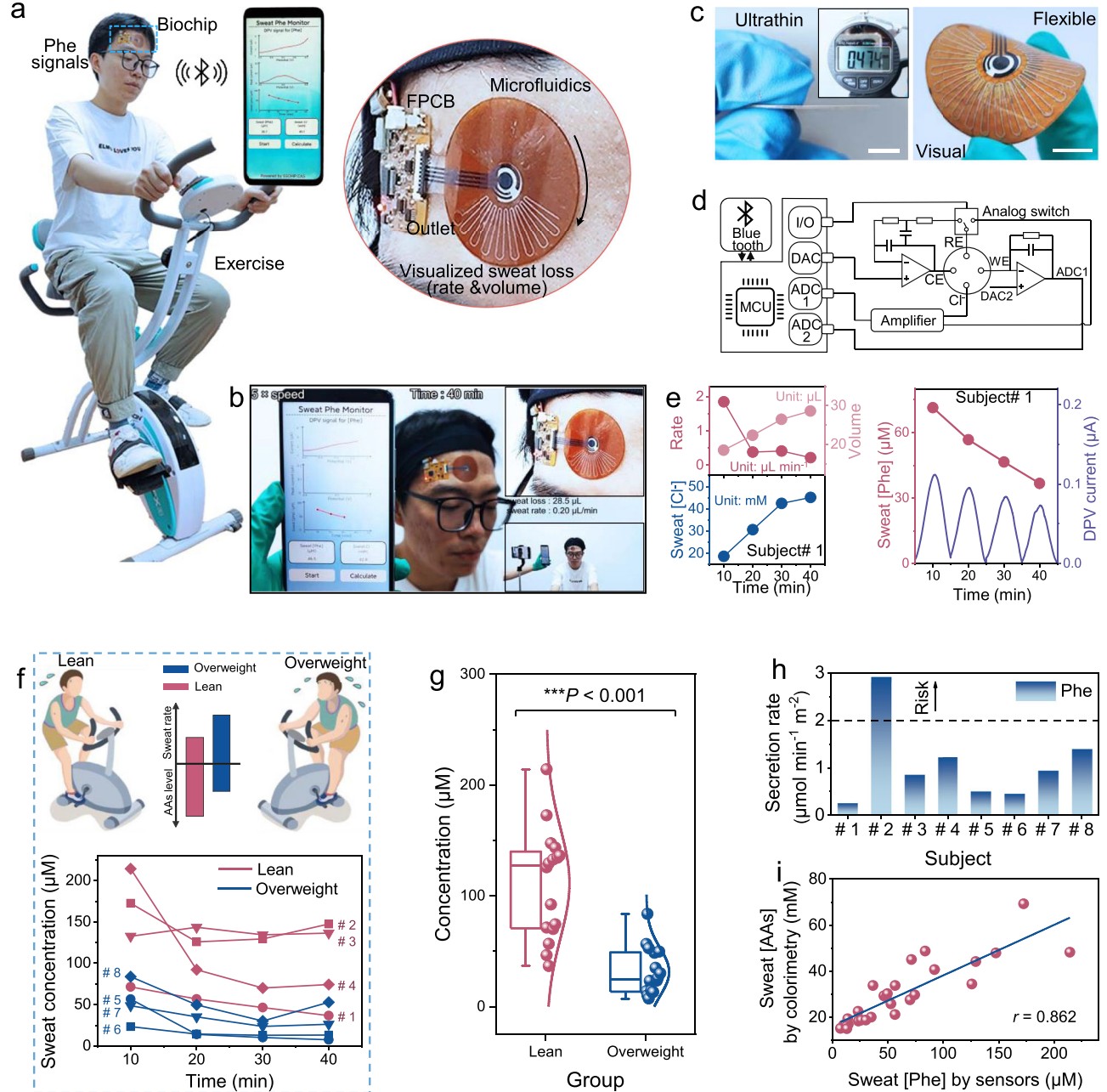

**Fig. 4 | On-body evaluation of wearable multimodal biochip for dynamic exercise sweat analysis and assessment. a** Photographs of a subject wearing the biochip on forehead and a smartphone app interface. **b** Screenshot of a frame of the sensor validation video. **c** Ultrathin and flexible demonstration of the sensor performance. Scale bar, 1 cm. **d** Hardware block diagram of the flexible circuit for the sweat sensor. **e** Real-time continuous monitoring of sweat loss (left top), sweat chloride (left bottom) and Phe concentrations (right) along with corresponding DPV data from 0.4 to 0.6 V per scan obtained from the forehead of subject #1. **f, g** Dynamic sweat Phe measurement (f) and corresponding box-and-whisker plot (g) in two groups of male subjects: lean/normal group ($n$ = 4 subjects) and overweight group ($n$ = 4 subjects). Difference in sweat Phe levels collected for two groups is statistically significant (two-tailed Wilcoxon rank-sum test, W = 378; ***$P$ < 0.001; $P$ = 0.00000169). The box ends represent the 25th and 75th percentiles. The horizontal line in each box represents the median. The upper and lower whiskers represent the maxima and minima, respectively, which refer to the range of non-outlier data values. **h** Sweat Phe secretion rates of eight subjects calculated by sweat rate and Phe concentration during 10 to 20 min of exercise. **i** High-positive correlation of sweat AAs and Phe levels for the validation of sweat Phe secretion rate as an indicator to reflect sweat AA loss for exercise-related metabolic risk assessment. The data are based on 28 sweat samples collected from the two groups (lack of data on Subject #3 due to his low forehead sweat volume). The solid line represents the linear-fitted trendline.

The matching flexible circuit can implement excitation DPV potential and differential OCP measurement as well as corresponding multimodal bio-signal retrieval, processing, and transmission (Fig. 4d). As a demonstration of the sensor validation (Supplementary Video 3), the multimodal sensor was placed on the forehead of a healthy volunteer (subject # 1) during a cycling exercise trial at moderate intensity

[around 80% maximal heart rate (HR$_{max}$)] for 40 min in a climate-controlled room (22 to 25 °C, 20 to 40% relative humidity).

The sweat rate was computed from the captured sweat volume and reached a maximum level during the initial stage of perspiration (10 min), and then decreased until stabilization (Fig. 4e). This trend stems from thermoregulation, adaption, and equilibration of the

body's response to the exercise load as expected[57,58]. Meanwhile, there was a rising trend in the sweat chloride concentrations during exercise. More importantly, the real-time Phe measurement also showed a decreasing trend (Fig. 4e) because the endogenous Phe loss from the plasma gradually dominates while the contribution from skin AAs diminishes[20]. Similar trends as mentioned above were also observed in a low intensity exercise of jogging (around 60% $HR_{max}$) with lower sweat rates (Supplementary Fig. 28). The simultaneous measurement of the above three types of sweat indictors demonstrates the capability of our integrated biochip for multimodal sweat sensing. Among them, Simultaneous measurement of Phe concentration and sweat rate aided a quantitative assessment of exercise-induced AA loss and thereby helps to identify individuals with exercise-related metabolic risk, which have relatively high Phe secretion rate (Fig. 4f)[20].

To demonstrate practical application, we selected eight male volunteers with different physiological characteristics from the previously recruited 16 subjects for monitoring sweat Phe levels under the same trial conditions as described above. An analogous decrease trend was observed in all subjects, while there was lower sweat Phe concentrations in overweight subjects (BMI ≥ 25) than in lean subjects (BMI < 25) according to BMI classification (Fig. 4f and Supplementary Table 3). A non-parametric analysis of variance (Wilcoxon rank-sum test) was performed for statistical comparison, revealing a significant difference between the two groups (W = 378; ***$P < 0.001$) (Fig. 4g). This difference supports that overweight people tend to have lower levels of sweat Phe than lean people due to the concentration dilution effect caused by excessive sweat volume/rate[59]. Combined with the previous negative correlation between sweat rate and Phe concentration (Fig. 1b), these findings suggest that the partitioning mechanism of sweat Phe may mainly be attribute to passive transport from interstitial fluid (ISF) or blood[15,36] in addition to from the skin itself.

Sweat Phe secretion rates of all subjects were calculating by measuring and multiplying sweat rates and Phe concentrations (Fig. 4h), which is an important and suitable indicator for assessing the level of AA loss during exercise without interindividual variability according to statistical analysis (Supplementary Fig. 29). In order to verify this approach, we determined the corresponding AA concentrations of all sweat samples in order to analyze the relation between sweat AA and Phe levels and obtained a high Pearson correlation coefficient of 0.862 (Fig. 4i). Therefore, as expected, the sweat Phe secretion rate can reliably indicate the sweat-facilitated loss of amino acids during exercise, which is useful for assessing exercise metabolic risk and for potentially guiding nutritional supplementation to address losses in sweat and to maintain nitrogen balance[20]. Owing to an appreciably larger sweat Phe secretion rate compared to other volunteers (Fig. 4h), subject # 2 was identified as an individual with high sweat-facilitated loss of AAs and a candidate for protein supplementation after exercise.

### Evaluation of sweat Phe sensor for diet management and serum correlation

As confirmed above and reported before[14,20], the levels of AAs in sweat is mainly attributed to ISF or blood partitioning as sweating progresses. Accordingly, beyond its practical use in exercise management, the sweat Phe sensor has additional potential applications to understand the correlation of Phe levels in sweat versus serum such as in the case of diet management (Fig. 5a). Although it has been reported that sweat AA levels are associated with blood AA levels[30,60], their metabolic correlation during exercise has not been well studied, especially for non-NMF AAs.

To evaluate the use of our sweat sensors for non-invasively assessing serum Phe levels, a pilot study was conducted for continuous sweat Phe monitoring and corresponding serum Phe quantification on representatives (Subjects # 1 and 5) of two BMI groups before and after protein intake (Fig. 5b–f). In each later stage of exercise sweating

(plateau after 30 min), protein diet intake resulted in elevated sweat Phe levels in both subjects, while decreased levels were measured after rest (Fig. 5b). Moreover, the successive decrease of Phe levels in the initial stage of exercise sweating (10 min) points to the consumption of skin Phe and its untimely replenishment (Supplementary Fig. 30). However, Phe levels measured after 20 min did not show this change trend. Combined with the above two different phenomena, it could be inferred that skin Phe in sweat is no longer dominant at this time, but is replaced by the contribution from blood. Importantly, there was a greater extent of Phe percentage fluctuation (especially for sweat changes) in the overweight subject than in the lean subject, which is likely due to different metabolic conditions (Supplementary Fig. 31).

In addition, there was also a strong positive correlation between sweat Phe and AA levels (Fig. 5c), demonstrating the potential of using our sweat sensor for diet management by assessing both changes in serum Phe levels and sweat AA loss. Here, the accuracy of the sweat Phe sensor for testing human sweat samples was validated by enzyme-linked immunosorbent assay (ELISA) using commercial Phe kits (Fig. 5d). Taking the plateau of sweat Phe level in exercise as a comparative index with corresponding serum Phe levels quantified by ultra-performance liquid chromatography with mass spectrometry (LC-MS) (Supplementary Fig. 32), good agreement in the level changes between sweat and serum Phe before and after protein intake was observed (Fig. 5e). High Pearson correlation coefficients of 0.878 (Subject #1) and 0.947 (Subject #5) were observed between sweat and serum Phe levels (Fig. 5f, top). However, the findings also suggested interindividual variability in the correlation between serum and sweat Phe levels due to the large difference in the fitted line. To reduce the interindividual variability, sweat Phe concentrations were normalized to the sweat Phe secretion rate by multiplying by the individual stabilized sweat rate during exercise. This normalized sweat Phe indicator showed a strong correlation with serum Phe levels while also having a similar slope of the fitted line between the two subjects (Fig. 5f, bottom). The difference in the line intercept is likely due to the difference in the physiological properties of subjects and Phe content on the skin surface. Furthermore, the positive interindividual correlation between sweat and serum Phe levels became stronger after sweat rate normalization (Supplementary Fig. 33). In short, by introducing sweat rates to reduce interindividual variability, sweat Phe secretion rates are a potentially suitable indicator to investigate the correlation between serum and sweat Phe. As such, the ability of our sensors to detect both sweat Phe levels and sweat rates displays a superior capacity to assess serum Phe levels over other available sensor options, and supports the feasibility of exploring phenylalanine as a sweat biomarker.

### Biocompatibility

Considering that the biochip may need to be worn for a long period of time in diet management applications, we also conducted a biocompatibility test of the biochip. As a representative skin cell line, human immortalized keratinocytes (HaCaT) were cultured on the biochip as the experimental group, whereas HaCaT was cultured in a blank medium as the control group. As shown in Fig. 5g, the fluorescence microscopy images show the survival status of the HaCaT cells after 2 or 4 days incubation. There was no significant reduction in relative cell viability of HaCaT cells on the biochip compared to the control group (Fig. 5h), indicating that the biochip is safe for prolonged wear and suitable for sweat multimodal detection in diet management.

## Discussion

We have presented an integrated wearable biochip for the multimodal detection of sweat Phe and chloride concentrations as well as sweat loss levels (volume and rate). A unique Phe sensing method is reported based on the Phe-imprinted enzyme-mimicking MIP that allows for direct electrocatalytic oxidation of Phe with high sensitivity and

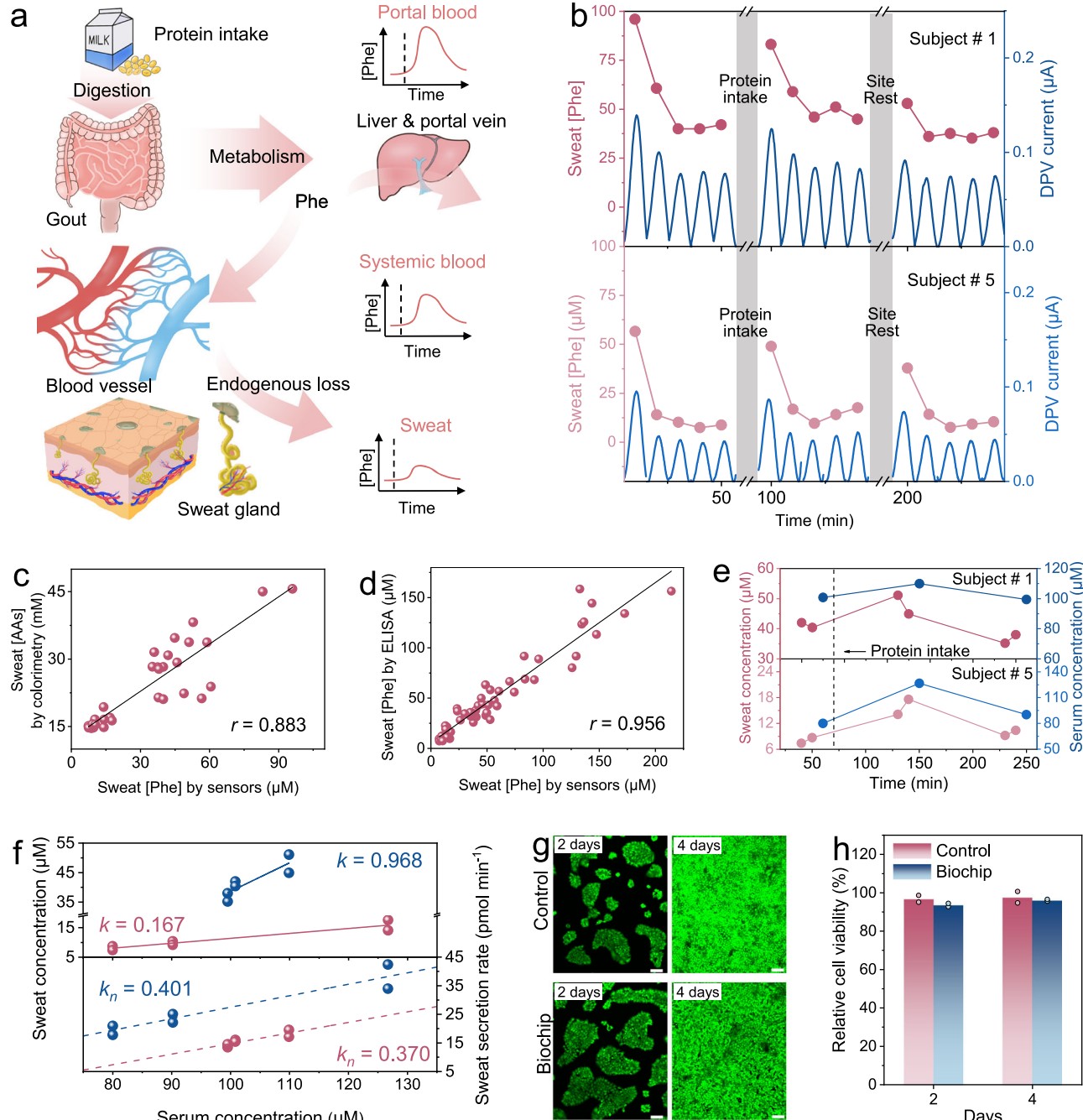

**Fig. 5 | In situ sweat Phe analysis for assessing serum levels and protein diet intake effects. a** Metabolic pathway of Phe along with serum and sweat Phe fluctuations caused by protein intake during exercise. **b** Dynamic changes of sweat Phe levels from two subjects with different BMI in three periods of exercise, including before and after protein diet intake as well as after rest. DPV data from 0.4 to 0.6 V. **c** Correlation between sweat AAs and Phe levels in the intake-exercise experiment from two different subjects. **d** Sensor-measured Phe concentrations in sweat samples versus corresponding ELISA readouts. Data was measured from all sweat samples collected in the above evaluation experiment. The solid line represents the linear-fitted trendline. **e** Comparative study of sweat and serum Phe levels in the three periods of exercise from subject #1 (top) and subject #5 (bottom). **f** Correlations between sweat and serum Phe levels before (top) and after (bottom) the sweat rate normalization from two different subjects. Lines represent the fitted trendlines. **g**, **h** Fluorescence microscopy images (**g**) and relative cell viability (**h**) of HaCaT cells after 2 or 4 days incubation in biocompatibility test (*n* = 2 independent experiments). Scale bar, 100 μm.

selectivity for sweat monitoring. Furthermore, an elaborate multi-purpose microfluidics was designed via a scalable and low-cost laser-engraving technique to achieve multiple goals, including sweat neutral pH buffering for stable Phe detection, rapid sweat sampling and refreshing for high temporal resolution, and sweat flow visualization for readable sweat loss. By integrating these two highly complementary and mutually beneficial approaches, the wearable system worn on the skin surface enables the continuous, reliable, and rapid detection of Phe concentrations along with the accessible and visual readout of sweat loss during exercise. Potential improvements in the system could focus on the development of continuous sweat loss readings through electrical methods for more diverse application scenarios.

To demonstrate the practical value of our well-designed multi-modal biochip, we have used our system to determine a negative correlation of sweat Phe levels and sweat rates among individuals, and

thus analyze the potential mechanism of Phe partitioning. Additionally, we have combined these two indicators as a new indicator, sweat Phe secretion rate, for quantitatively assessing the exercise metabolic risk of two populations with different BMI. Finally, we have conducted a pilot study to understand the metabolic correlation of Phe content in sweat versus serum before, during, and after protein diet intake at exercise, revealing their high and similar correlations in two subjects after sweat rate normalization. All these results demonstrate that simultaneous determination of sweat amino acid concentration and sweat rate can offer new and valuable insights into the assessment of metabolic status and corresponding blood levels. To overcome the limited subjects' data in the study, future work will include the investigation of more subjects with different metabolic conditions to expand on the proof-of-concept data herein and to build a stronger correlation of sweat and serum Phe levels by considering the sweat dilution effect. Such approaches will pave the way for a range of clinical application scenarios, such as non-invasive and personalized low-Phe diet management for PKU patients. More importantly, our finding that non-NMF AAs in sweat mainly stem from the endogenous loss of plasma AAs as sweating progresses can be extended to other species. These measurement capabilities can be utilized to study more sweat-serum AA correlations and their detailed partitioning mechanisms with the help of multimodal sweat sensing in follow-up studies.

## Methods

### Fabrication and preparation of the sweat sensor

A predesigned four-electrode array mask was patterned and engraved on the PET label paper with adhesive backing by an Yttrium Aluminum Garnet (YAG) laser cutter (Suzhou Inngu Laser, China). The laser-patterned mask was attached to a PET film with a thickness of 125 μm (Huanan Xiangcheng Technology, China), and the layers of Cr/Au (20/50 nm) was deposited on the PET film via thermal evaporation. To prepare the RE and the chloride sensor, Ag/AgCl ink (ALS, Japan) was first uniformly brushed onto the preset RE and chloride sensor zones, and then the mask attached on the PET film was stripped. After drying overnight in the ambient environment, the pure Ag/AgCl was used as the sensitive material of chloride sensor. As for the common RE, 4 μL cocktail that contained 50 mg NaCl, 79.1 mg polyvinyl butyral (PVB), 2 mg Poly (ethylene glycol)-block-poly(propylene glycol)-block-poly(ethylene glycol) (F127), and 0.2 mg MWCNTs in 1 mL methanol was drop-casted on Ag/AgCl at the RE zone.

The preparation process of the PANI-MIP sensor for sensing Phe is illustrated in Supplementary Fig. 4. Before the polymerization, the thermally evaporated gold electrode at the WE zone was drip-coated and cleaned with piranha solution ($H_2SO_4$: $H_2O_2$ = 7:3 v/v). The whole synthesis of the E-MIP electrode was carried out in a conventional three-electrode cell equipped with an electrochemical workstation CHI 760E (CH Instruments, USA). The polymerization solution was prepared by dissolving 10 mM L-Phe and 10 mM aniline into 0.01 M phosphate buffered saline (PBS) (pH = 7). Then, the electropolymerization was conducted by means of cyclic voltammetry from −0.4 to 0.9 V versus the standard calomel electrode (SCE) for four cycles at a scan rate of 50 mV/s. After polymerization, the electro-elution was accomplished by multi-potential steps (MPS) switched forty times alternately between 0.9 and −0.9 V (each potentiostatic duration of 100 s) versus SCE, which over-oxidizes/electro-degrades the PANI-based MIP matrix to extract templates and thus results in the E-MIP electrode with selective and electrocatalytically active binding sites. Subsequently, the resulting electrode was immersed into 1× PBS (pH 7.0) for repetitive DPV scans (same parameters as for Phe measurement, see below for details) until a stable response was obtained. For the preparation of NIP and E-NIP electrodes, the same preparation method was applied as MIP and E-MIP, excluding the Phe molecule as the template during the electro-polymerization step.

### Characterization of the sweat sensor

All the in vitro characterizations of prepared sensors were performed using an electrochemical workstation CHI760E in 10×PBS (pH 7.0). The response of the chloride sensor was characterized by open circuit potential (OCP) measurements with varied chloride concentrations. The DPV analysis was performed for the E-MIP-based Phe sensor and corresponding other electrodes for comparison at varied physiological Phe concentrations. The DPV parameters were: range from 0.4 to 0.7 V; incremental potential of 0.001 V; amplitude of 0.03 V; pulse width of 0.05 s; sampling width of 0.02 s; pulse period of 0.1 s; sensitivity of $1 \times 10^{-6}$ A/V. Before each DPV scan, a constant potential of 0.4 V was applied across the WE and RE for 2 s to promote the binding of negatively charged Phe to the electrode and suppress potential interferants and product buildup. The EIS measurement and parameters are detailed in Supplementary Note 1. The selectivity study of the Phe sensor was tested by adding interferents successively at physiologically relevant high concentrations. The pH dependence of Phe sensor response was studied by DPV scans in 10×PBS (pH 7.0) with different pH range from 11 to 3 adjusted by sodium hydroxide or hydrochloric acid, and measured by a precision pH meter (Sunne, China). All peak current readouts of the DPV scans were presented after a standard straight-line baseline correction.

The material morphology was characterized by FSEM (Hitachi High-Tech, Japan) and AFM (Bruker, USA). The Fourier transform infrared (FTIR) spectra was measured by using attenuated total reflection (ATR) in the infrared spectrophotometer (Tianjin Gangdong, China).

### Computational simulation methodology of molecular interactions

All quantum-chemical calculations were achieved using Gaussian 16 software package. Geometric structures of the target (Phe) and the pre-polymerization complexes were optimized by DFT using the B3LYP hybrid functional with Grimme's dispersion correction of D3 version (Becke-Johnson damping). See Supplementary Note 2 for details.

### Fabrication, assembly and characterization of the multipurpose microfluidics

The vertically assembled five layers of microfluidics were fully fabricated by laser engraving using the YAG laser cutter. All patterns needed to be laser-engraved were pre-designed using AutoCAD. From top to bottom, the top cover layer of transparent PET film (125 μm thick, thinner is also feasible) with the prepared sweat sensor was surface-roughened by the dot engraving mode to produce μm-scale etched dots (μ-dots). This surface roughening method by laser engraving was also applicable to other flexible or stretchable substrates (Supplementary Fig. 20). A thicker double-side tape (170 μm thick, 9495LE, 3 M, USA) was cut as the outflow channel layer with the chamber pattern and serpentine microchannel (240 μm width) to accommodate the volume of 1 μL per meander channel. The isolation layer of dark PI film (125 μm thick) was cut out the chamber to vertically interconnect the outflow and inflow microchannels through the chamber and provide a dark background for enhanced visualization (Supplementary Fig. 21). Then, a thinner double-side tape (100 μm thick, 7982, Crown, China) was cut as the inflow channel layer with elliptic inlets and the chamber pattern. Finally, skin adhesive medical dressing (Tegaderm, 3 M, USA) was incorporated and cut in the assembly process to form the skin interfacing layer with elliptic inlets. By cutting cross marks on each layer for alignment and positioning, the original 2D layers were vertically stacked and fluidically routed to form leakage-free, multi-layered, and 3D architectures. Here, the chamber for sweat reserving and sensing was formed by three layers (outflow channel, isolation, and inflow channel layers) all containing the chamber pattern. A

chamber-patterned filter paper (Whatman, USA) drip-coated and dried by 30 μL phosphate buffer solution (pH 7.5 to pH 7.8) was embedded within the chamber for neutral pH buffering.

For the characterization of multipurpose microfluidics, a syringe pump (Longer, China) was used and connected to a matching sized inlet of the microfluidics by a steel needle. The neutral pH buffering capability of embedded filter paper was evaluated by measuring the approximate pH range of the injected and buffered lactic acid solution (pH = 5) collected at the outlets via pH test papers (pH 5.4 to 7.0, Newstar, China). Flow rate measurement experiment was performed by comparing the constant injecting rates (0.5, 1, and 2 μL min$^{-1}$) to the calculated flow rates by dividing the readable filled volume to the time spent. Furthermore, a similar setup in the benchtop study was used to evaluate the ability of the wearable system integrating sweat sensors and microfluidics for accurate and stable wireless sensing of Phe. The measured DPV data was transmitted to a computer via Bluetooth for further baseline correction and data smoothing. To measure the optical property of visualized flow channel design, A UV/VIS spectrophotometer (Shimadzu, Japan) with an integrating sphere module enabled transmittance and reflectivity measurements of analogous chamber structures, including empty channels with or without μ-dots and water-filled channels with μ-dots (Fig. 3g and Supplementary Fig. 22).

### Numerical simulations of sweat dynamics in microfluidics

All simulations were carried out using COMSOL Multiphysics 6.0. For the verification of flow behavior under sweat sampling of the microfluidics with twelve inlets, a computational fluid dynamics module (Two-Phase Flow, Level Set interface) was used to simulate the water filling process with/without the filter paper (Porous Media Domain) in Supplementary Fig. 18. Moreover, as illustrated in Supplementary Fig. 19, the mass transport process was also simulated by coupling Transport of Diluted Species interface and Laminar Flow interfaces for the refreshing time analysis of microfluidics with/without the filter paper (Porous Media Domain). See Supplementary Note 3 for details.

### Wireless flexible circuit module and smartphone application

The wireless flexible circuit module implements excitation DPV potential waveform and differential OCP measurement as well as corresponding readout signals retrieval, processing, and transmission. As the core of the flexible circuit module, an STM32L15C8T6 ultra-low-power ARM Cortex-M3 32 MHz microcontroller (MCU) was programmed to facilitate system-level functionalities. For a DPV scan, the excitation potential waveform with the same previous parameters was applied across the WE and RE through a 12-bit digital-to-analog converter (DAC) built in MCU. A second-order low-pass RC filter further stabilized the potential of RE. The current response from the WE was extracted by a transfer impedance amplifier stage (TIA), which transferred the current signal to a voltage readout and then was read by a 12-bit analog-to-digital converter (ADC) built in MCU. The OCP measurement following the DPV scan was achieved by an AD8227ARMZ differential amplifier to effectively implement an instrumentation amplifier configuration. An analog switch was used to alternate the measurement mode of the flexible circuit.

The raw data was then wirelessly transmitted via Bluetooth (E104-BT5005A) and real-time displayed in a custom-developed smartphone APP. Before converting to presented concentration values, the raw DPV data was baseline-corrected, filtered, and smoothed on the APP to obtain reliable DPV peak current curves. Then, the in-situ concentration values were obtained and displayed in the user interface, along with recording the curve of Phe levels over time.

### Power source

The entire system was powered by a rechargeable 3.7 V lithium-ion polymer battery with desired capacity and size. A low dropout regulator (S-1206B33-M3T1G) was used to convert and produce a stable and separate 3.3 V digital and analog power supplies to serve the MCU and analog peripheral components, respectively, which creates separate digital and analog circuitry to prevent the digital circuitry from affecting analog signals.

### Human subject recruitment

The human subject experiments were conducted in compliance with the ethical standards of the institutional and/or national research committee and with the Declaration of Helsinki of the World Medical Association. This study was approved by the Ethical Committee of Tianjin Medical University General Hospital, Tianjin, China (Approval No. IRB2015-YX-009). The data were obtained with the informed consent of all participants. Each participant has been paid from 100 to 300 RMB (depending on the participation time) as the compensation for the test. The authors affirm that human research participants provided informed consent for the publication of the images in Fig. 4a, b, as well as the movies in Supplementary Videos 3, 4.

To avoid differences caused by gender and age, healthy male subjects between the ages of 23 and 27 were recruited through verbal recruitments. Informed consents were obtained from all study subjects before enrollment in the study. According to their BMI, the participating subjects were classified into two groups including a lean group with a BMI of 18.5 to 24.9 kg m$^{-2}$ and an overweight group with a BMI of 25 to 30 kg m$^{-2}$.

### On-body integrated system test

To validate the wearable multimodal sensor for dynamic exercise sweat analysis and metabolic assessment, a constant moderate-intensity cycling exercise with no additional human-participant risk was conducted on recruited volunteers, including four lean subjects and four overweight subjects (Supplementary Table 3). All subjects reported to the lab after overnight fasting. Their foreheads were cleaned with alcohol swabs removing skin contaminants (not limited to AAs) before the sensors were attached to the body. During cycling, a smartphone camera was used to take pictures of the sensor for subsequent analysis of the digital images for automatic sweat loss calculation. Moreover, the data from the sensors was wirelessly sent to the user interface via Bluetooth. Every ten minutes, we clicked the 'DPV' button in the smartphone APP to launch DPV scanning and OCP measurement for in situ quantification of Phe and chloride levels respectively in the subject's sweat at that moment. Meanwhile, sweat samples were also collected every 10 min from another area of the forehead of subjects using a funnel-shaped plastic bag. The sweat samples were then frozen at −25 °C in centrifuge tubes for further testing, such as the Phe and AAs colorimetric assay kits for sensor validation.

### Colorimetry for sweat sample analysis

For Phe standardized quantification, the L-Phenylalanine ELISA kit (Immusmol, France) was used for the competitive determination of Phe levels on collected sweat samples according to the manufacturer's instruction. The chromogenic reaction was monitored at 450 nm using a Multiskan SkyHigh microplate spectrophotometer (Thermo Fisher Scientific, USA) and the unknown concentrations of Phe was calculated by comparing their absorbance with a reference curve prepared with known standards. For AAs standardized quantification, the micro AA content assay kit (Solarbio, China) was used for the direct determination of AAs levels on collected sweat samples using ninhydrin derivatization, which forms a violet ninhydrin-amino acid complex. After mixing all reagents, standards, and samples thoroughly, the mixtures were incubated in a boiling water bath for 15 min and then cooled at a cold water bath for 5 min. The absorbance at a wavelength of 570 nm was measured using the microplate spectrophotometer, and utilized for the calculation of unknown AAs concentrations according to the

absorbance of a Glycine standard (10 mM). The above unspent reagents in the colorimetric assay kits were stored at 2–8 °C unless otherwise used.

## System validation with protein diet intake

To evaluate the system for non-invasively assessing serum Phe levels, a preliminary protein diet study was conducted on representative subjects from two BMI groups during exercise. The subjects reported to the lab after overnight fasting. The detailed human trial process is as follows: 1) a 50 min constant-load cycling exercise was performed on the subjects with the sweat analysis by the system from the forehead every 10 min, and the sweat samples were collected by the funnel-shaped plastic bag for further ELISA analysis. The sweat rate measured at this stage was used as the parameter to normalize the Phe concentration indicator in the subsequent sweat multimodal analysis; 2) Fresh capillary blood samples were immediately collected using a finger-prick approach. After cleaning the fingertip with an alcohol wipe and air drying, the fingertip was punctured with a custom-made blood-collection pen (Dunrse, China). Blood samples were collected with centrifuge tubes after the first drop of blood was wiped off. The collected blood samples were left at room temperature for 60 min and then centrifuged at 2000 for 15 min to separate the serum for LC-MS analysis; 3) The subjects were provided a standardized protein-rich snack, including a soy product and a pure milk. After diet intake, they are asked to sit and rest for 40 min to fully digest the intake and absorb the decomposed AAs; 4) Same as the first cycling, the subjects exercised for 50 min while sweat was analyzed by the wearable system and sampled for LC-MS analysis. Before sweat sensing via the previously used systems, the filter paper embedded in the microfluidic chamber was replaced with a new one, which was achieved by tearing off the old lower two layers of microfluidics and reassembling the new two layers. This replacement operation refreshes the pH buffering capacity of microfluidics to ensure the reliability of the Phe measurement in the next period without the need of a new sensor; 5) After the second cycling exercise, the subjects are asked to sit and rest for 40 min allowing the body to regulate the elevation of AAs levels caused by protein intake; 6) Same as the first and second cycling exercises as well as the replacement operation, the subjects performed the third exercise.

## LC-MS for serum analysis

The Phe concentration in the serum samples were determined by LC-MS with a multiple reaction monitoring (MRM) technique, which uses retention time as the main parameter used to identify analytes. Briefly, chromatographic separation was achieved using an ACQUITY UPLC I-Class system (Waters, USA) and Phe was separated using a UPLC HSS T3 column (Waters, USA) at a flow rate of 0.5 mL min$^{-1}$. Phe was detected using a triple quadrupole mass spectrometer XEVO TQ-S (Waters, USA) fitted with an electrospray ionization interface operated in positive ion mode. Serum Phe was quantified by the peak area using MassLynx according to the standard curve obtained from Phe standards at known concentrations.

## In vitro biocompatibility assessment

HaCaT (iCell Bioscience, China) were inoculated on a special medium for HaCaT at a density of $4 \times 10^4$ cells per pore, and cultured with both the control group and the sterilized biochip in an incubator at a constant temperature of 37 °C under 5% $CO_2$. The HaCaT were cultured in vitro for 48 h (2 days) and 96 h (4 days), washed one time with PBS. Half stained with 10 μL calcitonin-AM and 4.5 μL polyimide for 15 min, was used to observe the number of live/dead cells, and the other half incubated for another 4 h after adding 0.5 mL min$^{-1}$ MTT (Solarbio, China) was used to observe the relative cell activity by the microplate spectrophotometer.

## Statistical analysis

Origin 2018 was used for the linear fitting of corresponding data and the calculation Pearson correlation coefficient. SPSS Statistics 24 was used to assess the statistical significance. According to Shapiro–Wilk tests and Levene's tests respectively, the two sets of Phe data from the lean and overweight groups were conformed to normality and variance-heterogeneity. Therefore, Wilcoxon rank sum test (Mann–Whitney test) was used for comparison between two data groups.

## Reporting summary

Further information on research design is available in the Nature Portfolio Reporting Summary linked to this article.

## Data availability

All data supporting the findings of this study are available within the article and its supplementary files at https://doi.org/10.6084/m9.figshare.24786486. Any additional requests for information can be directed to, and will be fulfilled by, the corresponding authors. Source data are provided with this paper.

## Code availability

The codes used for simulation and data measurement are available from the corresponding author with upon reasonable request.

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

## Acknowledgements
The authors sincerely acknowledge financial support from the National Natural Science Foundation of China (NSFC Grant No. 62174152, 62374159, and 62022079) and from the Youth Innovation Promotion Association of Chinese Academy of Sciences (No. 2020115).

## Author contributions
B.Z., Z.L., and L.W., designed the research, B.Z. and L.W. wrote the paper, B.Z., X.Q., H.X., LC.L., L.L., ZX.L., Z.L., and L.W. performed the experiments, B. Z., L.L., and ZX.L. performed the first-principles calculations and simulations. B.Z., X.Q., J.A.J., N.-J.C., Z.L., and L.W. analyzed the data, B.Z. and L.C. designed the human studies. J.A.J., N.-J.C., Z. L. and L.W. revised the paper, Z.L. and L.W supervised the project. All authors substantially contributed to research and reviewed the manuscript.

## Competing interests
The authors declare no competing interests.
