## [Peer Review File · Nature Communications]

REVIEWER COMMENTS

Reviewer #1 (Remarks to the Author):

The authors reported a wearable sweat sensing system for wireless monitoring of multiple bio-signals in sweat, such as phenylalanine, chloride ion, and sweat loss. Based on the simultaneous measurement of phenylalanine concentration and sweat rate signals, their proposed sweat multimodal analysis performed some interesting and meaningful studies on the secretion mechanism of sweat phenylalanine and its correlation with serum levels. The measured and analyzed results demonstrated the practicability of the sweat sensing system in relevant applications. However, there are still some issues in this manuscript that need careful consideration and addressing by the authors. Acceptance of this manuscript is recommended after the following revisions:

1. Is the data in Fig. 1b actually measured? If it is used as a diagram corresponding to Fig. 1f, the diagram needs to be drawn according to the measured data. In other words, the points in Fig. 1b need to be distributed more discretely so that they fall within different risk areas. Moreover, a clear explanation about the corresponding relationship between secretion rate values and risk areas in Fig. 1f is necessary.
2. On page 11, the figure numbers annotated for Figs. 1f and 1g are incorrect.
3. Molecular imprinting behavior can be simply but effectively simulated by computing the molecular interaction and the bonding energies between a polymer monomer and a target. Why were the complex long chains of polyaniline chosen for the calculation of bonding energies between aniline and phenylalanine? The authors need to give specific reasons or advantages.
4. The intermolecular simulation didn't have detailed structural models of the PANI-E-NIP and PPY-E-MIP electrodes used for calculating the electron transfer between electrodes and phenylalanine.
5. The performance of electrochemical sensors also needs to be tested at varied temperatures and ionic strengths.
6. The reported electrochemical sensors had stable performance in neutral to weakly alkaline environments. Is it better to use a pH buffer filter paper loaded with weakly alkaline salts than the neutral one?
7. The authors measured the transmittance change to reflect the color change of the visualized flow channel before and after sweat filling. However, what is actually read here is the reflected light from the rough structure to the naked eye rather than the transmitted light. Therefore, it is recommended to measure the reflectivity change in Fig. 3g.
8. It contains a set of parallel lines with unknown meaning in Fig.4d. Please check carefully.
9. The developed sweat sensing system enables multimodal detection, including chloride ion concentrations, which are also important for exercise management. Therefore, the authors need to present relevant measurement data to reinforce the concept of multimodal sensing in this manuscript.

Reviewer #2 (Remarks to the Author):

Want et al. present a study on a wearable multimodal biochip designed to concurrently measure phenylalanine levels and sweat rates. Their proposed electrocatalytically active Molecularly Imprinted Polymer (MIP) serves as a sensing method, providing high sensitivity and selectivity. The authors establish a negative correlation between sweat phenylalanine levels and sweat rates, introducing a novel indicator, the Phe secretion rate. Additionally, they investigate the correlation of phenylalanine content in sweat versus serum.

The paper is written in a good structure, demonstrating clarity and accessibility. The novelty and contribution can be further developed. Additionally, I have several questions, particularly regarding the exercise tests, which I believe need clarification before publication.

1. In Fig 1f, the correlation between sweat rate and Phe level appears not very nice-related, particularly at slower sweat rates. Could the authors explain it and how you deal with the low rate?
2. The paper suggests that the system is designed for a continuous 40-minute exercise test. What are the implications or limitations after this duration?
3. The method of measuring sweat rate by naked eye in a visual manner raises questions about accuracy. Why not integrate a sweat rate sensor into the overall system for enhanced precision?
4. The data obtained from exercise tests are based on 40 minutes of continuous exercise. How would the system perform with intermittent pauses?
5. In lower sweat rate scenarios like walking or running, how does the system's performance compare to the current test conditions?
6. While the rationale behind considering sweat rate is clear, the methodology for studying it appears less accurate and comprehensive. Could the authors provide insights into potential improvements?

7. Regarding MIP sensing, it usually requires analytical removal to ensure the testing for the next time. How does this impact sensitivity and reliability in this work? I didn't see there was a step about it.

Reviewer #3 (Remarks to the Author):

This manuscript demonstrates a creative study on the detection and analysis of sweat phenylalanine (Phe) using a sweat multimodal biochip. The challenges and implications of sweat amino acid monitoring are systematically summarized in this manuscript. To overcome existing challenges, a series of novel sensing and analysis methods are proposed, including direct oxidation for Phe detection, multipurpose microfluidic design, and sweat rate normalization. In particular, the study of the interindividual correlation between sweat and blood Phe based on sweat rate normalization has a potential application prospect in healthcare, and can inspire other sweat biomarker analysis, not limited to amino acids.

Given the above reasons, I would like to recommend this work to be published in Nature Communications. Below are some suggestions for improvement.

1. The authors should explain the practical significance of tracking exercise metabolism and assessing its risk.
2. In Fig.1b and Fig. 1f, the authors should explain the basis for dividing high, medium, and low exercise metabolic risk areas.
3. Is the processing method of laser roughening the PET surface for visualizing flow channels suitable for other flexible substrates or even stretchable substrates?
4. Why is the pump flow rate associated with microfluidic testing in the range of 0.5 and 2.0 $\mu\text{L min}^{-1}$? Is there any reference basis for the value?
5. Does the hygroscopic expansion of the embedded filter paper in the detection chamber influence the electrochemical and colorimetric sensing?
6. In the study on the correlation between sweat and blood Phe, the authors should explain the basis for comparing sweat Phe concentrations after 20 minutes of exercise with serum Phe concentrations. How to define the time when the source of sweat phenylalanine from endogenous loss begins to dominate?
7. In Fig. 5, in addition to fitting the subjects' correlations individually, the authors should also fit and analyze the overall interindividual correlation between measured sweat and serum Phe levels before and after sweat rate normalization for a more comprehensive support in interindividual correlation improvement.

There are a few minor text and graphic errors that need to be corrected.

1. Page 11: Fig. 1i and Fig. 1j in the text are incorrect because there are no corresponding figures in Fig. 1, they should be Fig. 1f and Fig. 1g, respectively.
2. Page 14: Fig. 2f and Fig. 2g in the text are also incorrect, they should actually be Fig. 2h and Fig. 2i, respectively.
3. Some of the text in the figures of this manuscript are deviated or overlapping, including Fig. 1d and Supplementary Fig. 4.

Dear Reviewers,

Response to Reviewers for NCOMMS-23-50030-A

Thank you for your consideration of our manuscript, “**Interindividual- and Blood-Related Sweat Phenylalanine Multimodal Analytical Biochips for Tracking Exercise Metabolism**” for publication in *Nature Communications*. We also wish to thank the Reviewers for their insightful and helpful comments. Below, we have provided a detailed response to each item mentioned by the Reviewers, and indicate how we have incorporated their suggestions into the revision.

We would also like to summarize the responses to the Reviewers, as well as the updates made to the manuscript, as follows:

- 1) Extended Analysis and Improvement:** Per the Reviewers’ constructive suggestions, we have remeasured the subjects’ sweat loss (Fig. 1f, Fig. 1g, Fig. 3i, Fig. 4e, Fig. 4h, and Fig. 5f) in our study using the automatic reading algorithm to further improve accuracy based on the current system structure. Besides, we have carefully answered all other concerns of the Reviewers about system performance (reproducibility, Supplementary Fig. 14; swelling effect, Supplementary Fig. 17; reading comparison, Supplementary Fig. 24; temperature influence, Supplementary Fig. 26; other scenario, Supplementary Fig. 28) through reasonable analysis with measured data support.
- 2) Additional and Revised Discussions:** Per the Reviewer’s suggestions, we have added or revised explanations of our research basis, descriptions of our results, and their corresponding figures (Fig. 1b, Fig. 3g, and Fig. 4e, as well as Supplementary Fig. 9, Supplementary Fig. 22, Supplementary Fig. 27, Supplementary Fig. 29, and Supplementary Fig. 33) in order to better understand our study. With the reminders of the Reviewers, we have also revised some errors, such as format, ambiguous expression, and so on. We have highlighted all the revisions in red so that the Reviewers can comment on it after our revision.

Thank you again for your further consideration of the manuscript for publication in *Nature Communications*.

Best Regards,
Lili Wang, Professor

List of detail changes:

1. Page 4-5: An explanation of the practical significance for tracking exercise metabolism and assessing its risk has been added according to the reviewer's suggestion.
2. Fig. 1b and its caption has been revised according to the reviewer's suggestion.
3. Fig .1d has been corrected according to the reviewer's suggestion.
4. Page 11: An explanation for the not very nice correlation at slower sweat rates in Fig. 1f has been added according to the reviewer's suggestion.
5. Page 11: The basis for dividing the different risk areas in Figs. 1b and 1f has been explained in detail according to the reviewers' suggestions.
6. Page 15: The description of sensor reproducibility has been revised.
7. Fig. 3g and its caption has been updated with reflectivity measurements according to the reviewer's suggestion, and the original transmittance measurement has been moved into the supplementary information as new Supplementary Fig. 22.
8. Page 18-19: A description of the data measurement method in Fig. 3g has been added.
9. Page 19-20: A discussion on the influence of temperature and ionic strength on Phe sensors has been added.
10. Fig. 4d has been corrected according to the reviewer's suggestion.
11. Fig. 4e and its caption has been revised according to the reviewer's suggestion.
12. Page 20-21: A description of the trend in measured sweat chloride concentrations during exercise in new Fig. 4e has been added.
13. Page 21: An additional performance illustration of the system in low sweat rate scenarios according to new Fig. 4e have been added.
14. Page 21: A description for the system's multimodal sweat sensing capability according to new Fig. 4e has been added.
15. Page 25-26: The basis for comparing sweat Phe levels after 20 min of exercise with corresponding serum Phe levels has been added according to the reviewer's suggestion.

16. Page 26: The comparison of the overall interindividual correlation before and after sweat rate normalization has been added according to the reviewer's suggestion.
17. Page 28: An outlook for future system improvements has been added according to the reviewer's suggestion.
18. Page 32: A description of the universality of our surface roughening method for different substrates has been added according to the reviewer's suggestion.
19. Page 34: The description for the optical property method has been revised.
20. Page 9 in Supplementary Note 2: The conformers of the PANI-E-NIP and PPY-E-MIP electrodes have been described according to the reviewer's suggestion.
21. Page 10 in Supplementary Note 2: The description of simulated results has been revised.
22. Page 12 in Supplementary Note 3: The necessary for accurately calibrating the changed liquid capacity after the introduction of filter paper has been explained.
23. Page 16-17 in Supplementary Note 3: The introduced computer vision algorithm for automatic sweat loss reading and its calculation results compared with visual reading results have been described and analyzed.
24. Supplementary Fig. 4 has been corrected according to the reviewer's suggestion.
25. Supplementary Fig. 9 has been revised according to the reviewer's suggestion.
26. Supplementary Fig. 27 and Supplementary Fig. 29 have been revised.
27. According to the revisions, the following figures has been added and highlighted in red: Supplementary Figs. 14, 17, 20, 22, 24, 26, 28, and 32.
28. The list of references has been updated.

Reviewer Comments to Author:

Reviewer 1

Comments:

The authors reported a wearable sweat sensing system for wireless monitoring of multiple bio-signals in sweat, such as phenylalanine, chloride ion, and sweat loss. Based on the simultaneous measurement of phenylalanine concentration and sweat rate signals, their proposed sweat multimodal analysis performed some interesting and meaningful studies on the secretion mechanism of sweat phenylalanine and its correlation with serum levels. The measured and analyzed results demonstrated the practicability of the sweat sensing system in relevant applications. However, there are still some issues in this manuscript that need careful consideration and addressing by the authors. Acceptance of this manuscript is recommended after the following revisions:

Response: We appreciate the reviewer's interest in our studies for sweat phenylalanine multimodal analysis, and thank for the positive comments and constructive suggestions. Below, we respond individually to each specific point raised by the Reviewer and discuss how we have incorporated the reviewer's suggestions into the revision.

1. Is the data in Fig. 1b actually measured? If it is used as a diagram corresponding to Fig. 1f, the diagram needs to be drawn according to the measured data. In other words, the points in Fig. 1b need to be distributed more discretely so that they fall within different risk areas. Moreover, a clear explanation about the corresponding relationship between secretion rate values and risk areas in Fig. 1f is necessary.

Response: Thank you for the constructive remark. In this work, Fig. 1b was just used as a separate diagram to exhibit the negative correlation between sweat rates and sweat phenylalanine concentrations among individuals under an ideal condition, which indicated the sweat rate quantification plays an important role in discovering sweat biomarkers and exploring their health implications. Thus, the data in Figure 1 is a schematic diagram of the ideal state, not the actual measured data. Moreover, although Fig. 1f was a relatively discrete measured data under the influence of various external factors, the overall data conforms to a strong negative correlation (Pearson's correlation coefficient, $r = -0.713$).

Based on your valuable suggestion, we have adjusted the position of the risk areas divided in Fig. 1b in the revised manuscript to be closer to the schematic data, so that the data fall within different risk areas. Additionally, we have clearly explained the corresponding relationship between secretion rate values and risk areas as well as their division basis for a more comprehensive understanding in the revised manuscript (highlight in red).

Fig. 1 b, Indicative negative correlation between sweat Phe concentrations and sweat rates among individuals, along with the calculation method of Phe secretion rate (inlets area of 75.36 mm^2) and the division of different metabolic risk areas.

Revision:

(Page 12 in the revised manuscript)

“Here, we divided Fig. 1f into three metabolic risk areas, where lower than $1 \mu\text{mol min}^{-1} \text{ m}^{-2}$ or higher than $2 \mu\text{mol min}^{-1} \text{ m}^{-2}$ were defined as low risk or high risk, respectively, and between the two values was defined as medium risk. The division was based on a combination of actually measured values from subjects in this study and the similar sports science research²⁰.”

20. Dunstan R. H., et al. Sweat Facilitated Amino Acid Losses in Male Athletes during Exercise at 32-34 degrees C. *PLoS One* 11, e0167844 (2016).

2. On page 11, the figure numbers annotated for Figs. 1f and 1g are incorrect.

Response: Thank you for the remark. We have corrected the figure numbers annotated for Fig. 1f and Fig. 1g in the revised manuscript.

3. Molecular imprinting behavior can be simply but effectively simulated by computing the molecular interaction and the bonding energies between a polymer monomer and a target. Why were the complex long chains of polyaniline chosen for the calculation of bonding energies between aniline and phenylalanine? The authors need to give specific reasons or advantages.

Response: Thank you for the constructive remark. We agree that the molecular interaction simulation between the polymer monomer and the target molecular were commonly used in molecular imprinting techniques, and can explain the selectivity of molecular imprinted polymers^{1,2}. However, using long polymer chains for intermolecular simulation is a more efficient approach than the polymer monomer in the following aspects^{3,4}:

1. The results of calculated binding energy based on the long chain are more realistic and accurate than those based on the monomers, because the target

molecular is bound to the polymer chain rather than the monomer;

2. The simulated structural changes of the polymer chain as well as their binding forms and energy changes could effectively and realistically reflect the binding and elution process between the polymer matrix and the target molecular in the imprinting behavior. This cannot be achieved by simulations based on the monomer. The simulation of the molecular imprinting process in this work have been highlighted in red on Page 9 in the revised supplementary information.
3. The simulated polymer long chains with different could represent their respective actual electrodes and were used in the next electrochemical behavior simulation for calculating the electron transfer between electrodes and phenylalanine in this work.

The above reasons motivated us to use the long chains of polyaniline for the simulation instead of the commonly-used simple aniline monomer. Accordingly, we have added an explanation about the choice of long polyaniline chains for molecular interaction simulations in Supplementary Note 2 (highlight in red).

1. BelBruno J. J. Molecularly Imprinted Polymers. *Chem. Rev.* 119, 94-119 (2019).
2. Mukasa D., et al. A Computationally Assisted Approach for Designing Wearable Biosensors toward Non-invasive Personalized Molecular Analysis. *Adv. Mater.* 35, e2212161 (2023).
3. Tang W., et al. Touch - Based Stressless Cortisol Sensing. *Adv. Mater.* 33, 2008465 (2021).
4. Bagdžiūnas G. Theoretical design of molecularly imprinted polymers based on polyaniline and polypyrrole for detection of tryptophan. *Mol. Syst. Des. Eng.* 5, 1504-1512 (2020).

Revision:

(Page 8 in the revised supplementary information)

“Here, the long polymer chains were used for accurate calculations of binding energies, efficient simulation of the molecular imprinting process, and further electrochemical behavior simulation.”

4. The intermolecular simulation didn't have detailed structural models of the PANI-E-NIP and PPY-E-MIP electrodes used for calculating the electron transfer between electrodes and phenylalanine.

Response: Thank you for the constructive remark. We apologize for our omission of these structural models that have been calculated previously by intermolecular simulation. In the revised supplementary information, we have added these conformers (PPY_re_Phe and nPANI_ov_Phe) in new Supplementary Fig. 9, which have been used to mimic the PPY-E-MIIP and PANI-E-NIP respectively.

With the reviewer's comment, we have also a corresponding description in

Supplementary Note 2 (highlight in red). Moreover, the description related to new Supplementary Fig. 9 has also been revised and highlighted in red.

Supplementary Fig. 9 Intermolecular simulation and electron transfer calculation. **a**, Values of electron transfer (Δq) between five different electrodes and Phe w/ or w/o an external electric field. **b**, absorption behavior of Phe on the gold electrode. **c-g**, Electrochemical behavior and electron transfer of Phe on the gold (**c**), PPY (**d**), E-MIP (**e**), E-NIP (**f**), and MIP (**g**) electrodes. The red arrow represents the direction of the external electric field.

Revisions:

(Page 9 in the revised supplementary information)

“A single three units with alternating benzenoid and quinoid rings (nPANI_{ov}_Phe) was chosen to mimic the PANI-E-NIP electrode due to its fewer binding sites compared to the PANI-E-MIP electrode. In addition, the conformer (PPY_{re}_Phe) used for mimicking PPY-E-MIP electrode was constructed based on the relevant reference¹².”

(Page 10 in the revised supplementary information)

“Importantly, the E-MIP electrode exhibits the greatest Δq of 3.924e, followed by the E-NIP electrode with Δq of 3.013e, the PPY electrode with Δq of 2.861e, the gold electrode with Δq of 2.757e, and the MIP electrode with smallest Δq of 0.103e (Supplementary Fig. 9c-g).”

5. The performance of electrochemical sensors also needs to be tested at varied temperatures and ionic strengths.

Response: Thank you for the constructive remark. We have prepared a batch of electrochemical phenylalanine sensors for testing the performance at varied temperatures. As shown in new Supplementary Fig. 26 below, the results showed that the response of peak current of our sensors were stable in the skin physiological temperature range (25 to 39 °C), whose peak current variation does not exceed 10%.

Additionally, the test results in Figs. 3k, l could demonstrate the performance changes of the phenylalanine sensor at varied ionic strengths (see Page 19 highlighted in red in the revised manuscript). Specifically, by continuously injecting a fixed concentration of sweat samples into the integrated system, we simulated the ionic strength changes and pH changes in the sensing chamber due to the dilution of the original pH-buffering salt by sweat. A stable response (< 10% variation) of the phenylalanine sensor was been obtained within the injected 80 μ L sweat samples. The results indicated that the introduction of pH-buffering filter paper in microfluidics can effectively alleviate the influence of pH and ionic strength changes on the sensor response within the design expectation.

With the reviewer's suggestion, we have added new Supplementary Fig. 26 in the revised supplementary information, and the following discussion on the influence of temperature and ionic strength on phenylalanine sensors in the revised manuscript.

Supplementary Fig. 26 Performance of the Phe sensor at varied temperatures.

Revision:

(Page 19-20 in the revised manuscript)

“The results shown that the embedded pH-buffering filter paper in microfluidics can effectively alleviate the influence of pH as well as ionic strength changes on the sensor response within the design expectation. Furthermore, A stable response of the sensor was been obtained in the skin physiological temperature range (Supplementary Fig. 26).”

6. The reported electrochemical sensors had stable performance in neutral to weakly alkaline environments. Is it better to use a pH buffer filter paper loaded with weakly alkaline salts than the neutral one?

Response: Thank you for the constructive remark. We agree with the possibility of using an alkaline pH-buffering filter paper to replace the role of neutral one in this work. However, the chloride sensing electrode and the reference electrode based on Ag/AgCl in the integrated system are unstable in alkaline solution environment, which will react into black silver oxide in use, causing the electrode potential to drift. Hence, considering the stability of other electrodes, we used the neutral pH-buffering filter paper rather than the alkaline one in this work.

7. The authors measured the transmittance change to reflect the color change of the visualized flow channel before and after sweat filling. However, what is actually read here is the reflected light from the rough structure to the naked eye rather than the transmitted light. Therefore, it is recommended to measure the reflectivity change in Fig. 3g.

Response: Thank you for the constructive suggestion. We strongly agree that it is better to measure the reflectivity change in the visualized outflow channel than the transmittance change. Therefore, by using the UV/VIS spectrophotometer with an integrating sphere module, we have added reflectivity measurements of channels in three different situations, including the empty channels with and without μ -dots as well as the filled channel with μ -dots. As shown in new Fig. 3g below, there is a large difference in visible light reflectivity between flow channels with and without μ -dots, while this difference decreases significantly after sweat fills the channel with μ -dots. This phenomenon of change in optical properties is consistent with that measured by the transmittance.

With the reviewer's suggestion, we have used new Fig. 3g in the revised manuscript to replace the original one, which have been moved into the revised supplementary information as new Supplementary Fig. 22. We have also revised and highlighted in red related description for the optical measurement in the revised manuscript.

Fig. 3 g, Optical reflectivity of empty channels with or without μ -dots and filled

channels with μ -dots.

Supplementary Fig. 22 Optical transmittance of empty channels with or without μ -dots and filled channels with μ -dots.

Revision:

(Page 34 in the revised manuscript)

“A UV/VIS spectrophotometer (Shimadzu, Japan) with an integrating sphere module enabled transmittance and reflectivity measurements of analogous chamber structures, including empty channels with or without μ -dots and water-filled channels with μ -dots (Fig. 3g and Supplementary Fig. 22).”

8. It contains a set of parallel lines with unknown meaning in Fig. 4d. Please check carefully.

Response: Thank you for the remark. We have deleted the meaningless and confusing parallel lines in Fig. 4d in the revised manuscript.

Fig. 4 d, Hardware block diagram of the flexible circuit for the sweat sensor.

9. The developed sweat sensing system enables multimodal detection, including chloride ion concentrations, which are also important for exercise management. Therefore, the authors need to present relevant measurement data to reinforce the concept of multimodal sensing in this manuscript.

Response: Thank you for the constructive remark and suggestion. We have

added the chloride concentration change during exercise measured by the sweat multimodal system in new Fig. 4e according to the data from Supplementary Video 1. The three types of measured data in new Fig. 4e shows and reinforces the concept of multimodal sensing in this work, including sweat loss (sweat volume and rate), sweat chloride and phenylalanine concentrations. Moreover, the following descriptions about new Fig. 4e were added and highlight in red in the revised manuscript.

Fig. 4e, Real-time continuous monitoring of sweat loss (left top), sweat chloride (left bottom) and Phe concentrations (right) along with corresponding DPV data from 0.4 to 0.6 V per scan obtained from the forehead of subject #1.

Revisions:

(Page 22 in the revised manuscript)

“Meanwhile, there was a rising trend in the sweat chloride concentrations during exercise.”

(Page 22 in the revised manuscript)

“The simultaneous measurement of the above three types of sweat indicators demonstrates the capability of our integrated biochip for multimodal sweat sensing.”

Reviewer 2

Wang et al. present a study on a wearable multimodal biochip designed to concurrently measure phenylalanine levels and sweat rates. Their proposed electrocatalytically active Molecularly Imprinted Polymer (MIP) serves as a sensing method, providing high sensitivity and selectivity. The authors establish a negative correlation between sweat phenylalanine levels and sweat rates, introducing a novel indicator, the Phe secretion rate. Additionally, they investigate the correlation of phenylalanine content in sweat versus serum.

The paper is written in a good structure, demonstrating clarity and accessibility. The novelty and contribution can be further developed. Additionally, I have several questions, particularly regarding the exercise tests, which I believe need clarification before publication.

Response: We thank the reviewer for the positive and insightful comments. Below, we respond individually to each specific point raised by the Reviewer and discuss how we have incorporated the reviewer's suggestions into the revision.

1. In Fig 1f, the correlation between sweat rate and Phe level appears not very nice-related, particularly at slower sweat rates. Could the authors explain it and how you deal with the low rate?

Response: Thank you for the insightful remark. The not very nice relationship between sweat rate and Phe level is due to the different skin qualities or conditions of the subjects, which is particularly noticeable in people with low sweat rates. As we mentioned on Page 3 in Introduction (Highlight in red) and related references proposed^{1,2}, the amino acids in sweat, including Phe, arise from both the blood plasma and the skin surface. Phe content on the skin surface has a great relationship with the skin quality of each person³, which will inevitably affect sweat Phe levels measured by biosensors, especially for people with low sweat rates. For these individuals, although the Phe contribution from the skin surface decreases through dilution as exercise duration, this source is difficult to be exhausted through perspiration due to their low sweat volumes/rates. This effect resulted in the relatively poor relationship in the low sweat rate region in Fig. 1f, and was not obvious in the high sweat rate region. Nevertheless, this effect does not affect their metabolic risk classification due to their low sweat rates. This phenomenon is a good discovery that was not mentioned in the manuscript. Thank the reviewer again for this profound and insightful suggestion for improvement.

With the reviewer's suggestion, we have added an explanation for the above phenomenon in Fig. 1f in the revised manuscript (highlight in red).

1. Mark H. & Harding C. R. Amino acid composition, including key derivatives of eccrine sweat: potential biomarkers of certain atopic skin conditions. *Int. J. Cosmet. Sci.* **35**, 163-168 (2013).
2. Dunstan R. H., *et al.* Sweat Facilitated Amino Acid Losses in Male Athletes during Exercise at 32-34 degrees C. *PLoS One* **11**, e0167844 (2016).
3. Watabe A., *et al.* Sweat constitutes several natural moisturizing factors, lactate, urea, sodium, and potassium. *J. Dermatol. Sci.* **72**, 177-182 (2013).

Revision:

(Page 11 in the revised manuscript)

"Remarkably, the correlation in the low sweat rate region was poorer than that in the high sweat rate region because the skin surface Phe content in sweat, which is affected by individual skin quality differences, was difficult to be

exhausted by perspiration at low sweat rates²⁰.”

20. Dunstan R. H., *et al.* Sweat Facilitated Amino Acid Losses in Male Athletes during Exercise at 32-34 degrees C. *PLoS One* **11**, e0167844 (2016).

2. The paper suggests that the system is designed for a continuous 40-minute exercise test. What are the implications or limitations after this duration?

Response: Thank you for the insightful remark. The 40-minutes test time was the effective detection time of our system during intense exercise with a persistently high sweat rate of $2 \mu\text{L min}^{-1}$. In effect, according to our test results in the forehead (a body part with high sweat rates) of subjects, our system could provide more than 50 minutes of detection time due to a decline in sweat rate as exercise duration. This detection time was longer in other body parts with lower sweat rates.

In addition, our work also considered the requirement for long-term exercise scenarios and successfully applied the system for longer periods of sweat monitoring. As highlighted in red on Page 39-40 in the manuscript, we can open the microfluidic chamber and then replace the filter paper with a new one for the subsequent measurement with unchanged performance. This approach was mentioned for the study of sweat-serum Phe correlations where the system enabled three successive sweat monitoring lasting over 200 minutes. Thank again for the reviewer's valuable comment.

3. The method of measuring sweat rate by naked eye in a visual manner raises questions about accuracy. Why not integrate a sweat rate sensor into the overall system for enhanced precision?

Response: Thank you for the insightful remark. We also appreciate your suggestions for improvement regarding the measurement accuracy of our sweat loss sensor. Yes, generally, the reading based on the naked eye is inferior to the automatic reading in terms of accuracy and error control. In our work, in order to reduce the error gap and improve the accuracy, we designed the elongated, visualized, and meandering serpentine outflow microchannel with a volume reading scale of $0.5 \mu\text{L}$, which is sufficient for μL -level sweat quantification here. Taking the lowest actually measured sweat rate of $0.3 \mu\text{L}/\text{min}$ during multimodal analysis at the forehead as an example, the user could read the color change in one visual volume scale in about 2 minutes (corresponding to a sweat volume of $0.5 \mu\text{L}$) with an error of about 16.67%, and then calculate the measured sweat rate of $0.25 \mu\text{L}/\text{min}$ with an error of about 16.67%. In fact, our study on sweat phenylalanine multimodal analysis did not require such a low reading time interval. The longer the reading interval, the smaller the error in our sweat quantification. The limited extension of the reading time interval does not affect the relative value of sweat rates as well as its trend of sweat rate over time during exercise. A relatively long reading time interval between 5 and 10 minutes for sweat rate quantification could decrease

or even circumvent measurement errors and satisfy the requirement of multimodal analysis. Such long reading time intervals have been commonly used in existing field measurement studies for exercise sweat analysis by using the regional absorbent patch method^{4,5} and the colorimetric microfluidic method⁶⁻⁸. Therefore, our colorimetric sweat loss sensor is sufficient and suitable in terms of accuracy and error control for our study on sweat phenylalanine multimodal analysis.

With your encouraging and insightful suggestion, we have integrated the function of automatic sweat loss reading into the terminal based on the computer vision algorithm for further improvement in accuracy and error control based on the current colorimetric structure. The algorithm is based on the spatial color distribution and three-dimensional patch geometry to achieve more accurate quantification of sweat loss from original snapshots of the microfluidic patch on the forehead of eight subjects⁵. Based on this, we have remeasured the subjects' sweat loss in our study using the automatic algorithm, including the sweat rates in Figs. 1f, 3i, and 4e and the calculated Phe secretion rates in Figs. 1g, 4h, and 5f as well as their related figures and tables in the revised supplementary information. The data in the above figures in our work have been refreshed with the remeasured results. We have also provided relevant explanations in the revised manuscript to avoid confusion caused by revisions.

Additionally, we have also compared the sweat loss remeasured by the automatic algorithm with that visually read by users' naked eyes. As shown in new Supplementary Fig. 24 in the revised supplementary information, the visually measured results are highly positively correlated with the algorithm results (Supplementary Figs. 24a, b). Calculated by a two-tailed t-test (Supplementary Fig. 24c), there is no significant difference between the two results ($P = 0.370$). The above results indicate that although the readings based on naked eyes may introduce measurement errors, they are within the allowable range of error and have no impact on the analysis conclusion.

Supplementary Fig. 24 Comparison of the sweat loss values calculated from visual reading and algorithm. a,b, High-positively correlations between two methods for measuring sweat volumes (a) and sweat rates (b). **c**, Line sequence diagram of sweat rates measured by two methods. Statistical

analysis based on a two-tailed t-test showed that there was no significant difference between these two groups of paired data (n = 8).

4. Buono M. J., Ball K. D. & Kolkhorst F. W. Sodium ion concentration vs. sweat rate relationship in humans. *J. Appl. Physiol.* **103**, 990-994 (2007).
5. Baker L. B. Sweating Rate and Sweat Sodium Concentration in Athletes: A Review of Methodology and Intra/Interindividual Variability. *Sports Med.* **47**, 111-128 (2017).
6. Baker L. B., *et al.* Sweating Rate and Sweat Chloride Concentration of Elite Male Basketball Players Measured With a Wearable Microfluidic Device Versus the Standard Absorbent Patch Method. *Int. J. Sport. Nutr. Exerc. Metab.* **32**, 342-349 (2022).
7. Baker L. B., *et al.* Skin-Interfaced Microfluidic System with Machine Learning - Enabled Image Processing of Sweat Biomarkers in Remote Settings. *Adv. Mater. Technol.* **7**, 2200249 (2022).
8. Baker L. B., *et al.* Skin-interfaced microfluidic system with personalized sweating rate and sweat chloride analytics for sports science applications. *Sci. Adv.* **6**, eabe3929 (2020).

Revisions:

(Page 16-17 in the revised supplementary information)

“Computer vision algorithm for automatic sweat loss reading and comparative analysis

The photos used in the algorithm to automatically calculate sweat loss came from original snapshots of the system patch on the foreheads of eight subjects over a period of 10 to 20 minutes during exercise. All photos were converted to RAW format for eliminating artifacts. The algorithm based on traditional computer vision techniques identified the locations of relevant features, including the geometric outline and orientation of the microfluidic patch as well as the location of color-swath variations of the visualized outflow channel within the outline. Sweat volumes could be determined by locating the interface between the white microchannels that have not changed to the background color and the background-color microchannels due to sweat filling. Sweat rates could be computed from sweat volumes and the time interval of picture capturing.

To verified the accuracy of colorimetric sweat loss measurement with distinct visual color change, the sweat volume and rate measured by the automatic algorithm are compared to that visually read by users' naked eyes (Supplementary Fig. 24). The visually measured results are highly positively correlated with the algorithm results (Supplementary Fig. 24a, b). Calculated by a two-tailed t-test (Supplementary Fig. 24c), there is no significant difference between the two results (P = 0.370). Therefore, without the assistance of the automatic reading function, the sweat loss sensor with 0.5 μ L volume resolution developed for this work can also measure accurate readings with naked eyes.” (Page 18-19 in the revised manuscript)

“Additionally, a computer vision algorithm was developed for automatic sweat loss reading (Supplementary Note 3).”

4. The data obtained from exercise tests are based on 40 minutes of continuous exercise. How would the system perform with intermittent pauses?

Response: Thank you for the insightful remark. The APP-controlled sweat Phe and chloride levels detection developed in our system is performed on demand, and the electrochemical measurement only runs after the user clicks the “Start” button on the APP. For the sweat rate measurement, intermittent exercise pauses will not stop sweating due to hysteresis of heat dissipation, and the body is still sweating to reduce the core temperature. Therefore, the system measures sweat indicators as usual during exercise with intermittent pauses.

5. In lower sweat rate scenarios like walking or running, how does the system's performance compare to the current test conditions?

Response: Thank you for the insightful comment. It can be inferred from the measured data in Fig. 3m that the decrease in sweat rate will not affect the accuracy and reliability of electrochemical phenylalanine measurement. Importantly, according to your good suggestion, we have used our system to monitor the sweat Phe concentrations and sweat loss information of Subject# 1 during jogging [around 60% maximal heart rate (HR_{max}) for 50 min]. As shown in new Supplementary Fig. S28, the subject had lower sweat rates during low-intensity jogging compared to the original moderate intensity cycling exercise, which required longer time to fill the sensing chamber and then enable the system for sweat indicators measurement. Furthermore, the real-time Phe measurement has also shown a similar decreasing trend (new Supplementary Fig. S28b). In summary, our system can perform normally and accurately in lower sweat rate scenarios.

With the reviewer’s suggestion, we have added the system's performance in the lower sweat rate scenario of jogging in the revised supplementary information (Supplementary Fig. S28), and an additional related description in the revised manuscript (highlight in red).

Supplementary Fig. 28 Real-time sweat monitoring during jogging. a, sweat loss (volume and rate) measured by the automatic reading algorithm. **b,**

Phe concentrations along with corresponding DPV data from 0.4 to 0.6 V per scan obtained from the forehead of subject #1.

Revision:

(Page 22 in the revised manuscript)

“Similar trends as mentioned above were also observed in a low intensity exercise of jogging (around 60% HR_{max}) with lower sweat rates (Supplementary Fig. 28).”

6. While the rationale behind considering sweat rate is clear, the methodology for studying it appears less accurate and comprehensive. Could the authors provide insights into potential improvements?

Response: Thank you for the insightful remark. The accuracy and comprehensiveness of sweat loss measurement in our system can be improved in the future in two aspects. First, in terms of optimizing the current system structure, the image recognition function based the above computer vision algorithm for automatic sweat loss quantification can be integrated into the current interactive APP. All sweat indicators can be measured through the optimized APP, which improve the system’s integration and measurement comprehensiveness at the software level. Second, in the future exploration of other sweat metabolites, the current colorimetric sweat loss measurement can also be optimized into an electrical method based on a pair of continuous electrodes to meet the need of continuous quantification with short reading time intervals in lower sweat rate scenarios. However, the increased costs associated with these improvements also need to be carefully trade off.

With the reviewer’s comment, we have added a future outlook for system improvements in Discussion (highlight in red).

Revision:

(Page 28 in the revised manuscript)

“Potential improvements in the system could focus on the development of continuous sweat loss readings through electrical methods for more diverse application scenarios.”

7. Regarding MIP sensing, it usually requires analytical removal to ensure the testing for the next time. How does this impact sensitivity and reliability in this work? I didn’t see there was a step about it.

Response: Thank you for the insightful remark. The common MIPs act as ‘artificial antibodies’⁹ that requires an additional regeneration step to remove bound target molecules for the next detection. Unlike the above regular MIPs, the MIP electrode we developed is more like an ‘artificial enzyme’ that enables the direct electrocatalytic oxidation of Phe according to our experimental and simulation results. During the sensing process, the selectively bound Phe molecules are reacted under the external electric field, and the products are

subsequently separated from the binding sites due to structural mismatch. Therefore, our method of MIP sensing does not require analytical removal to ensure the next testing.

As shown in new Supplementary Fig. 14, we have tested the response change of an E-MIP electrode for six continuous successive measurements in a solution with a fixed Phe concentration to provide data support for sensor reproducibility. Moreover, we have revised the related description in the revised manuscript (highlight in red).

Supplementary Fig. 14 Reproducibility of the Phe sensor. a,b, DPV scans (a) and variation (b) of six continuous successive measurements using an E-MIP electrode in presence of 200 µM Phe.

9. Wang M., *et al.* A wearable electrochemical biosensor for the monitoring of metabolites and nutrients. *Nat. Biomed. Eng.* **6**, 1225-1235 (2022).

Revision:

(Page 15 in the revised manuscript)

“The all electro-processed MIP layer could be formed scalably on evaporated gold electrodes, which resulted in high reproducibility in terms of batch-to-batch consistency and continuous successive measurement stability (Fig. 2k and Supplementary Fig. 14).”

Reviewer 3

This manuscript demonstrates a creative study on the detection and analysis of sweat phenylalanine (Phe) using a sweat multimodal biochip. The challenges and implications of sweat amino acid monitoring are systematically summarized in this manuscript. To overcome existing challenges, a series of novel sensing and analysis methods are proposed, including direct oxidation for Phe detection, multipurpose microfluidic design, and sweat rate normalization. In particular, the study of the interindividual correlation between sweat and blood Phe based on sweat

rate normalization has a potential application prospect in healthcare, and can inspire other sweat biomarker analysis, not limited to amino acids. Given the above reasons, I would like to recommend this work to be published in Nature Communications. Below are some suggestions for improvement.

Response: We thank the reviewer for the positive comments and valuable suggestions. Below, we respond individually to each specific point raised by the Reviewer and discuss how we have incorporated the reviewer's suggestions into the revised manuscript.

1. The authors should explain the practical significance of tracking exercise metabolism and assessing its risk.

Response: Thank you for the valuable remark. It is important to track the loss of various metabolites during exercise and assess metabolic risk based on monitored data. Taking amino acids as an example, exercise would increase the requirement for protein intake and turnover in order to offset the increased losses of free amino acids arising from demands for energy metabolism, muscle tissue anabolism, and sweat excretory losses^{1,2}. When protein intake is unavailable, excessive sweat amino acids loss caused by long-term exercise could exacerbate the requirement for proteolysis and arise a negative nitrogen balance, which further leads to muscle wasting and damage. Therefore, continuous monitoring of sweat amino acid loss during exercise can effectively assess the metabolic risk associated with excessive free amino acid losses and avoid fatigue and pain.

With the reviewer's suggestion, we have added a discussion to explain the practical significance of tracking exercise metabolism and assessing its risk in Introduction (highlight in red).

1. Dunstan R. H., *et al.* Diverse characteristics of the urinary excretion of amino acids in humans and the use of amino acid supplementation to reduce fatigue and sub-health in adults. *Nutr. J.* **16**, 19 (2017).
2. Dunstan R. H., *et al.* Sex differences in amino acids lost via sweating could lead to differential susceptibilities to disturbances in nitrogen balance and collagen turnover. *Amino Acids* **49**, 1337-1345 (2017).

Revision:

(Page 4-5 in the revised manuscript)

"In this regard, tracking exercise metabolism and assessing risk by monitoring sweat AA losses can avoid a net negative nitrogen balance due to depletion of free AAs resources, which leads to fatigue and pain during exercise^{20,35}."

Additional of reference

35. Dunstan R. H., *et al.* Diverse characteristics of the urinary excretion of amino acids in humans and the use of amino acid supplementation to reduce fatigue and sub-health in adults. *Nutr. J.* **16**, 19 (2017).

2. In Fig.1b and Fig. 1f, the authors should explain the basis for dividing high, medium, and low exercise metabolic risk areas.

Response: Thank you for the valuable remark. We apologize for forgetting to provide relevant division basis before. According to a professional sports science field research³, the athletes studied were divided into three distinct group characterized by their sweat facilitated loss of amino acids (SFLAA): (a) the “Low” group showed the highest sweat loss volume and the lowest sweat amino acids concentration; (b) the “High” group showed the lowest sweat loss volume but the highest sweat amino acids concentration; (b) the “intermediate” group’s values fell between the above two group. The inverse correlation between sweat volume and sweat amino acids concentration among people in this group classification is similar to the negative correlation between sweat rate and sweat Phe concentration as shown in Fig. 1f in this work. Among athletes, the “high” group had the highest quantity of amino acid lost through sweat. Therefore, those individuals in the “high” group faced a greater risk of amino acid depletion (related to muscle wasting and damage) in body during exercise than other groups. Based on this conclusion with convincing arguments, we combined the data characteristics of sweat Phe secretion rates measured in this work to divide the risk areas in Fig.1b and Fig. 1f.

With the reviewer’s suggestion, we have added a discussion to explain the basis for dividing high, medium, and low exercise metabolic risk areas in this work (highlight in red).

3. Dunstan R. H., *et al.* Sweat Facilitated Amino Acid Losses in Male Athletes during Exercise at 32-34 degrees C. *PLoS One* **11**, e0167844 (2016).

Revision:

(Page 12 in the revised manuscript)

“Here, we divided Fig. 1f into three metabolic risk areas (corresponding to Fig. 1b), where lower than $1 \mu\text{mol min}^{-1} \text{m}^{-2}$ or higher than $2 \mu\text{mol min}^{-1} \text{m}^{-2}$ were defined as low risk or high risk, respectively, and between the two values was defined as medium risk. The division was based on a combination of actually measured values from subjects in this study and the similar sports science research²⁰.”

3. Is the processing method of laser roughening the PET surface for visualizing flow channels suitable for other flexible substrates or even stretchable substrates?

Response: Thank you for the valuable remark. We have roughened the surfaces of different flexible substrates, including polyimide (PI) and stretchable polydimethylsiloxane (PDMS), through the laser dot engraving mode. As shown in new Supplementary Fig. 20, the same effect as PET have been observed on both substrates, whose optical reflectivity change due to surface roughening.

With the reviewer's insightful suggestion, we have added new Supplementary Fig. 20 in the revised supplementary information, and a related description in the revised manuscript (highlight in red).

Supplementary Fig. 20 Photographs of different flexible substrates after surface roughening by laser engraving.

Revision:

(Page 33 in the revised manuscript)

“This surface roughening method by laser engraving was also applicable to other flexible or stretchable substrates (Supplementary Fig. 20).”

4. Why is the pump flow rate associated with microfluidic testing in the range of 0.5 and 2.0 $\mu\text{L min}^{-1}$? Is there any reference basis for the value?

Response: Thank you for the valuable remark. Generally, the sweat loss rate of human body is in the range of 200-2000 $\text{g m}^{-2} \text{h}^{-1}$ during exercise sweating^{4,5}. Based on the inlets area of 75.36 mm^2 and the sweat density of 1 g mL^{-1} , the injection flow rate of the syringe pump should be in the range of 0.25-2.5 $\mu\text{L min}^{-1}$ (see Page 12 highlighted in red in the revised supplementary information). Accordingly, the pump flow rate was set in the range of 0.5-2.0 $\mu\text{L min}^{-1}$ corresponding to sweat rates in the range of 400-1600 $\text{g m}^{-2} \text{h}^{-1}$ during moderate-intensity exercise⁶ set in this work. Most of the sweat rates we measured by the multimodal system fall within this range.

4. Zhong B., Jiang K., Wang L. & Shen G. Wearable Sweat Loss Measuring Devices: From the Role of Sweat Loss to Advanced Mechanisms and Designs. *Adv. Sci.* **9**, e2103257 (2022).

5. Taylor N. A. & Machado-Moreira C. A. Regional variations in transepidermal water loss, eccrine sweat gland density, sweat secretion rates and electrolyte composition in resting and exercising humans. *Extreme Physiol. Med.* **2**, 4 (2013).

6. Smith C. J. & Havenith G. Body mapping of sweating patterns in male athletes in mild exercise-induced hyperthermia. *Eur. J. Appl. Physiol.* **111**, 1391-1404 (2011).

5. Does the hygroscopic expansion of the embedded filter paper in the detection chamber influence the electrochemical and colorimetric sensing?

Response: Thank you for the valuable remark. The hygroscopic swelling of the embedded filter paper will not affect the accuracy of electrochemical and colorimetric sensing. Specifically, in Fig. 1e, we obtained a negligible change in the performance between the electrochemical sensor with or without the embedded filter paper (see Page 11 highlighted in red in the revised manuscript), which indicates that the introduction of filter paper, including its effect of hygroscopic swelling, does not affect the electrochemical sensing. Meanwhile, when calculating and pre-calibrating the fluid capacity (16 μL) of the microfluidic chamber embedded with a filter paper, we considered volume change caused by the occupancy and hygroscopic swelling of filter paper (see Page 12 highlighted in red in the revised supplementary information). We have also measured the thickness change of filter paper before and after the hygroscopic swelling. As shown in new Supplementary Fig. 17 below, the thickness only changed by $\sim 20\%$ after swelling.

With the reviewer's comment, we have added new Supplementary Fig. 17 and the following discussion in Supplementary information (Highlight in red).

Supplementary Fig. 17 Thickness change of the filter paper before and after hygroscopic swelling (n = 4).

Revision:

(Page 12 in the revised supplementary information)

“After the introduction of filter paper, it is important to accurately calibrate the changed liquid capacity by considering the increase in the swelling thickness of the filter paper (Supplementary Fig. 17) and its intrinsic water absorption volume for subsequent colorimetric sweat loss sensing.”

6. In the study on the correlation between sweat and blood Phe, the authors should explain the basis for comparing sweat Phe concentrations after 20 minutes of exercise with serum Phe concentrations. How to define the time when the source of sweat phenylalanine from endogenous loss begins to dominate?

Response: Thank you for the valuable remark. For correlation analysis with serum Phe concentrations, the choice of the time period of corresponding sweat Phe concentration was based on the fact that the sweat Phe content arises from both the blood plasma and the skin surface, among which the contribution from plasma would dominate as exercise duration due to the dilution of skin Phe by sweat^{3,7,8} (Highlighted in red on Page 3 in the revised manuscript). In the study shown in Fig. 5, it can be observed that the sweat Phe concentrations measured during each exercise of all subjects decreased first but remained stable after 20 minutes of exercise (Fig. 5b). This uniform trend in sweat Phe concentrations across multiple exercises in all subjects is consistent with the above fact. Moreover, the sweat Phe concentrations measured in the first 10 minutes decreased with the increase in the number of exercises, which indicated the consumption of skin Phe and its untimely replenishment (Highlighted in red on Page 25 in the manuscript). However, the concentrations measured after 20 min of exercise did not show this change (Fig. 5b). The above two different phenomena together reveal that skin Phe in sweat is no longer dominant at this time (20 min), but is replaced by the contribution from blood. In short, according to the above analysis and data support, we defined the time (20 min) when the source of sweat phenylalanine from blood endogenous loss begins to dominate, and thus compared the sweat Phe concentrations after 20 min of exercise with the corresponding serum Phe concentrations for correlation analysis.

With the reviewer's valuable comment, we have added a related description about Fig. 5 for a clearer explanation for defining the time when the source of sweat phenylalanine from endogenous loss begins to dominate.

3. Dunstan R. H., *et al.* Sweat Facilitated Amino Acid Losses in Male Athletes during Exercise at 32-34 degrees C. *PLoS One* **11**, e0167844 (2016).

7. Baker L. B. & Wolfe A. S. Physiological mechanisms determining eccrine sweat composition. *Eur. J. Appl. Physiol.* **120**, 719-752 (2020).

8. Murphy G. R., *et al.* Relationships between electrolyte and amino acid compositions in sweat during exercise suggest a role for amino acids and K⁺ in reabsorption of Na⁺ and Cl⁻ from sweat. *PLoS One* **14**, e0223381 (2019).

Revision:

(Page 25-26 in the revised manuscript)

"However, Phe levels measured after 20 min did not show this change trend. Combined with the above two different phenomena, it could be inferred that skin Phe in sweat is no longer dominant at this time, but is replaced by the contribution from blood."

7. In Fig. 5, in addition to fitting the subjects' correlations individually, the authors should also fit and analyze the overall interindividual correlation between measured sweat and serum Phe levels before and after sweat

rate normalization for a more comprehensive support in interindividual correlation improvement.

Response: Thank you for the valuable remark. We have fitted the overall interindividual correlations between sweat and serum Phe concentrations before and after sweat rate normalization in two subjects. As shown in new Supplementary Fig. 33 below, a more positive sweat-serum correlation (Supplementary Fig. 33a) has been found after normalizing sweat Phe concentrations with sweat rates than the original one (Supplementary Fig. 33b), indicating that sweat rate normalization for sweat Phe concentrations could improve the interindividual correlation by reducing inter-variability.

With the reviewer's suggestion, we have added new Supplementary Fig. 33 in the revised supplementary information and a related explanation to comprehensively support the effectiveness of our sweat rate normalization in improving interindividual correlation.

Supplementary Fig. 33 Interindividual correlations between sweat and serum Phe levels before (a) and after (b) sweat rate normalization in two subjects.

Revision:

(Page 27 in the revised manuscript)

“Furthermore, the positive interindividual correlation between sweat and serum Phe levels became stronger after sweat rate normalization (Supplementary Fig. 33).”

8. There are a few minor text and graphic errors that need to be corrected.

(1) Page 11: Fig. 1i and Fig. 1j in the text are incorrect because there are no corresponding figures in Fig. 1, they should be Fig. 1f and Fig. 1g, respectively.

(2) Page 14: Fig. 2f and Fig. 2g in the text are also incorrect, they should actually be Fig. 2h and Fig. 2i, respectively.

(3) Some of the text in the figures of this manuscript are deviated or overlapping, including Fig. 1d and Supplementary Fig. 4.

Response: Thank you for the remarks. We have corrected the above errors accordingly in the revised manuscript and the revised supplementary information.

Fig. 1 Schematics of wearable multimodal biochip for assessing exercise metabolic risk and serum correlation through sweat analysis. a, Schematic of the biochip on skin for multimodal sweat sensing and two sources of sweat Phe, including skin surface and blood partitioning. **b**, Indicative negative correlation between sweat Phe concentrations and sweat rates among individuals, along with the calculation method of Phe secretion rate (inlets area of 75.36 mm²) and the division of different metabolic risk areas. **c**, Mechanism of the E-MIP electrode for detecting Phe by direct electrocatalytic oxidation and theoretical simulation of charge transfer between the E-MIP electrode and a Phe molecule under an external electric field. The red arrow represents the direction of the electric field. **d**, Working principle and cross-section of the microfluidic module with vertically assembled structure. **e**, Comparison of Phe DPV responses of the integrated wireless system and pristine electrodes. **f**, Correlation of sweat rates and Phe levels among individuals at the same exercise time point, along with division of different sweat Phe secretion rates. The starred data indicates the subject with the highest secretion rate in this study. Data from 16 healthy subjects. **g**, Correlations of sweat and serum Phe level in two subjects. The dashed lines represent linear-fitted trendlines. **h**, Comparison of recent advances in sweat sensing. **: this work; Tyr: tyrosine; BCAAs: branched chain amino acids; CRP: C-reactive protein; HB: β-hydroxybutyrate; UA: uric acid; VC: Vitamin C.

Supplementary Fig. 4 Schematic of the preparation procedure and sensing mechanism of the E-MIP-based Phe sensor.

REVIEWERS' COMMENTS

Reviewer #1 (Remarks to the Author):

The current edition of the revision can be accepted as it is.

Reviewer #2 (Remarks to the Author):

Thanks for the revision and response. All of my previous concerns have been addressed.

Reviewer #3 (Remarks to the Author):

The author has solved the problem I am concerned about very well. I recommend this work to be published in Nature Communications.

Reviewer Comments to Author:

Reviewer #1

Comments: The current edition of the revision can be accepted as it is.

Response: We greatly thank the Reviewer #1 for the positive comment and the acknowledgment on our work. We also appreciate the constructive remarks and suggestions throughout the revision.

Reviewer #2

Comments: Thanks for the revision and response. All of my previous concerns have been addressed.

Response: We are very grateful to the Reviewer #2 for the insightful remarks and suggestions in improving our work. We also appreciate your time and efforts in reviewing our work.

Reviewer #3

Comments: The author has solved the problem I am concerned about very well. I recommend this work to be published in Nature Communications.

Response: We thank the Reviewer #3 so much for the positive comment and valuable suggestions. We also appreciate your recommendation for the publication of our work.